# FORGING BETTER REWARDS: A MULTI-AGENT LLM FRAMEWORK FOR AUTOMATED REWARD EVOLUTION

## ABSTRACT

Large Language Models (LLMs) have shown increased autonomy in performing complex tasks, but the inference latency and fine-tuning cost impose significant limitations for their application in dynamic, real-time environments such as robotics and gaming. Reinforcement learning (RL), by contrast, offers efficient execution and has shown strong results in diverse domains. Yet its progress is often bottlenecked by the challenge of designing effective reward functions, which are typically sparse and require heavy manual effort to engineer. Recent work has explored LLM-based reward generation, reducing manual effort yet remaining unstable, unstructured, and opaque. Building on the enhanced reasoning capabilities of modern LLMs, we advance this line of research toward full automation by introducing structured reward initialization, evolutionary refinement, and explicit complexity modeling. These innovations reduce reliance on manual trial-and-error while enabling more stable, interpretable, and scalable reward design. We unify them into FORGE (**F**eedback-**O**ptimized **R**eward **G**eneration and **E**volution), a multi-agent framework that automatically forges increasingly effective reward functions. Extensive experiments across three games and a robotics task demonstrate the effectiveness of FORGE, achieving up to **38.5%** improvement over Eureka and **19.0%** over REvolve in the Humanoid task, while maintaining competitive token efficiency.

## 1 INTRODUCTION

Large language models (LLMs) and Vision Language Action (VLA) models have advanced rapidly, opening unprecedented opportunities for embodied intelligence (Andreas, 2022; Brohan et al., 2022; Schick et al., 2023; Ma et al., 2024b; Yang et al., 2024). These models excel at high-level reasoning, planning, and decision-making, enabling agents to interpret complex instructions and adapt to diverse environments. However, if every low-level action is delegated to LLM outputs, the resulting latency and inference cost become prohibitively high, rendering such approaches impractical for scalable deployment (Kaddour et al., 2023; Wang et al., 2024; Zhou et al., 2024). Humans address this challenge by combining deep, deliberate reasoning with fast, automatic motor execution—leveraging both reflective cognition and muscle memory in everyday tasks. This analogy suggests a promising paradigm for embodied AI: integrating the high-level reasoning of foundation models with the low-level efficiency of reinforcement learning (RL) (Yu et al., 2023; Xie et al., 2024; Sun et al., 2025).

Reinforcement learning (RL) (Sutton & Barto, 2018) naturally fills this role by providing low-level execution that is both efficient and scalable. RL agents have achieved remarkable performance in domains such as dexterous robotic control and locomotion, where once-trained policies can be deployed with negligible inference cost (Brohan et al., 2023; Kwon et al., 2023). Yet, this efficiency hinges critically on the design of reward functions that guide learning. Constructing such rewards is notoriously difficult: human designers must translate abstract objectives into precise signals, a process that often requires extensive trial-and-error and domain expertise (Chentanez et al., 2004; Yu et al., 2023). A recent survey reports that 92% of RL practitioners rely on manual reward tuning and 89% acknowledge their designed rewards to be suboptimal, frequently leading to unintended behaviors (Hadfield-Menell et al., 2020; Booth et al., 2023). Consequently, while RL excels at execution, the bottleneck of reward design makes the training pipeline costly and limits the broader adoption of RL in complex, real-world settings.

To alleviate this bottleneck, recent work has explored using LLMs to automate reward design. Methods such as Eureka (Ma et al., 2024a) and REvolve (Hazra et al., 2025) reduce manual effort by prompting LLMs to generate reward functions, lowering the burden of human engineering. While these approaches mark important progress, they remain far from fully automatic or reliable. Directly sampling executable functions from LLMs often produces unstable rewards that fail to generalize across environments. Moreover, the lack of structured mechanisms to organize and refine generated rewards leads to redundancy and ineffective candidates, wasting both training time and compute. Finally, without a principled measure of reward complexity, these methods offer little interpretability, making it difficult to analyze how reward design quality correlates with agent performance. As a result, existing approaches still suffer from instability, redundancy, and opacity in reward shaping.

We introduce FORGE (**F**eedback-**O**ptimized **R**eward **GE**neration), a multi-agent LLM framework that incrementally forges better reward functions through structured initialization and evolutionary refinement. The name reflects its central idea: rewards are not discovered in a single step but are repeatedly shaped and improved, much like metal forged under heat and pressure. Unlike prior approaches that directly sample functions from LLMs, FORGE begins with a Planner agent that reasons over task objectives, environment abstraction, and function interfaces to generate structured reward specifications. These specifications are then implemented as modular functions, providing strong zero-shot performance even before any refinement. Building on this foundation, FORGE applies an evolutionary process where reward functions are iteratively selected and combined under LLM guidance, enabling exploration beyond fixed encodings. Finally, a reward pool serves as specialized memory, while a depth measure quantifies structural complexity, ensuring stability and interpretability throughout the evolution. Together, these components establish a robust pipeline for automated reward evolution.

We validate FORGE through extensive experiments spanning both discrete and continuous control domains. Specifically, we evaluate on three classic gaming environments—Tetris, Snake, and Flappy Bird—as well as the continuous robotics benchmark Humanoid (MuJoCo). Across all tasks, we compare against strong baselines, including Eureka, REvolve, context-aware LLMs, and native sparse rewards. The results highlight three consistent trends: (i) the Planner initialization yields strong zero-shot performance, outperforming direct LLM sampling; (ii) the evolutionary refinement steadily improves reward quality, driving stable performance gains across environments; and (iii) the reward pool and depth measure enable interpretable analysis of reward complexity, revealing correlations between structural depth and agent performance. Importantly, FORGE achieves these gains without incurring higher token costs, demonstrating strong token efficiency relative to prior multi-agent LLM frameworks.

In a nutshell, our contribution can be summarized as follows.

- We propose FORGE, a new paradigm for automated reward evolution that combines planner-based initialization with evolutionary refinement, providing a structured and scalable framework for reward design.

- We introduce a reward pool and depth measure that jointly ensure stability, interpretability, and token efficiency, enabling reliable evolution without additional inference cost.

- Extensive experiments across four environments demonstrate the effectiveness of FORGE; in the challenging *Humanoid* task, it achieves up to **38.5%** improvement over Eureka and **19.0%** over REvolve in final rewards, while maintaining competitive token efficiency.

## 2 RELATED WORKS

**Reward Shaping.** Reward shaping is a persistent challenge in RL that traditionally requires significant domain expertise to craft precise reward. Algorithmic approaches that design intrinsic rewards with bonus-driven exploration emerges as potential solutions (Bellemare et al., 2016; Tang et al., 2017; Devidze et al., 2022), but are not easily generalizable to new environments. Inverse reinforcement learning approaches infer reward functions from demonstration but require human expertise and extensive data collection (Abbeel & Ng, 2004; Hadfield-Menell et al., 2020; Snoswell et al., 2020). Recent works that utilize LLM as reward designer, leveraging its reasoning ability and parametric knowledge to automate reward shaping over complex tasks (Yu et al., 2023; Ma et al., 2024a; Xie et al., 2024). Although they have achieved significant success, incorporating LLM as a primary

Figure 1: Pipeline of FORGE. The **Planner** agent reasons over the environment's objective, dynamics, and interface to produce textual reward specifications, which are converted into executable functions and stored in the **Reward Pool**. The **Engineer** agent then iteratively samples from the pool, trains policies, evaluates outcomes, and invokes the language model to refine reward functions, forming a continual loop of generation and selection.

decision-maker can produce unstable result and lead to performance degradation. In contrast, FORGE aims to automatically generate consistently improving reward functions in different environments.

**LLM-Based Autonomous Agents.** Autonomous agents powered by LLMs have demonstrated capabilities beyond textual conversation (Luo et al., 2025). Large collaborative agentic frameworks can deliver end-to-end products and conduct independent research without human intervention (Hong et al., 2024; Qian et al., 2024; Schmidgall et al., 2025; Singh et al., 2025). Yet the absences of efficient feedback mechanism present difficulty to improving existing solutions. Flexible agents with the capacity to navigate internet can respond to user request with up-to-date information, but performs poorly on specialized tasks due to lack of interactions with the environments (Yang et al., 2023; OpenAI, 2025a; Yang et al., 2025).

## 3 METHOD

We introduce FORGE, an LLM-based multi-agent framework that automates reward shaping for reinforcement learning. By iteratively generating and refining surrogate rewards, FORGE enables effective policy learning in environments where the extrinsic reward $\bar{R}$ is sparse or delayed.

**Preliminaries.** The objective of reinforcement learning is to learn a policy $\pi$ that maximizes the expected cumulative return

$$J(\pi, \bar{R}) = \mathbb{E}[J(\tau, \bar{R}) \mid \pi], \tag{1}$$

where $\tau$ denotes a trajectory and $\bar{R} : \mathcal{S} \times \mathcal{A} \to \mathbb{R}$ is the extrinsic reward provided by the environment. Since $\bar{R}$ often provides limited guidance in complex tasks, we introduce surrogate rewards $R$ that serve as alternative training signals for policies. Each surrogate $R$ is evaluated by the extrinsic return it induces:

$$J(\pi_R, \bar{R}) \approx \frac{1}{N} \sum_{i=1}^{N} \sum_{t=0}^{H} \bar{R}(s_t^{(i)}, a_t^{(i)}), \tag{2}$$

where $\pi_R$ is the policy optimized under $R$ and $N$ is the number of evaluation episodes. These surrogate rewards form the basis of our framework and will be iteratively refined through evolution.

**Method Roadmap.** Our framework, illustrated in Figure 1, proceeds in two stages. First, a planner agent generates an initial reward population from structured specifications, providing strong zero-shot performance (subsection 3.1). Second, rewards are iteratively refined through LLM-guided selection and crossover (subsection 3.2).

### 3.1 REWARD POPULATION SAMPLING

A central novelty of our approach lies in how the initial rewards are generated. Unlike prior methods such as Eureka (Ma et al., 2024a) and REvolve (Hazra et al., 2025), which directly prompt LLMs to output executable reward functions, we employ a *planner* agent (see Figure 1). The planner explicitly reasons over task objectives, environment dynamics, and the interaction interface to produce a textual

*reward specification.* This specification is then integrated with environment information and passed to the language model, which synthesizes modular reward components.

The *planner* agent then implements these components into executable functions. To standardize the implementation, we enforce a fixed function interface specifying input arguments and return type, while the agent completes the function body. The initialized population is as follow,

$$\mathbb{B} = \{R_1^{(0)}, R_2^{(0)}, \ldots, R_k^{(0)}\}, \quad \text{where} \quad R_i^{(0)} \sim \mathbb{P}_\theta(\cdot \mid p_{R_i}). \tag{3}$$

This design offers two key advantages. First, the modularized structure enables parallel training of policies under different reward candidates. Second, adapting to a new environment requires only editing the prompt $p_E$, without additional coding. Together, these properties ensure that our framework achieves strong performance even in zero-shot settings, providing a robust initialization for subsequent evolutionary refinement.

**Zero-shot Effectiveness.** Thanks to the planner's structured specification and modularized implementation, our framework achieves competitive performance even before any evolutionary refinement. This strong initialization distinguishes it from prior approaches that rely solely on direct LLM sampling, and serves as the foundation for the evolutionary stage.

Having established the initialization stage, we now describe how these rewards are organized and further developed.

**Reward Function Pool as Specialized Memory.** The planner-sampled rewards constitute the zeroth generation of the population (with depth set to 0). To manage and refine these candidates, FORGE maintains a reward function pool that serves as a specialized memory. The pool records all generated rewards together with their scores, preserving strong candidates while discarding weaker ones. Unlike context-aware LLM approaches that rely on raw history data and repeated inference, this curated pool allows efficient reuse of high-quality functions at no additional cost, while also providing a structured basis for subsequent evolution.

**Complexity Measure: Depth.** Building on the pool structure, we further introduce a measure of structural complexity to characterize how rewards evolve over generations. Each planner-sampled reward is initialized with depth $d = 0$. When a new reward $R'$ is created by combining two parent rewards $R_i^{d_i}$ and $R_j^{d_j}$ with depths $d_i$ and $d_j$ respectively , its depth is defined as

$$\text{Dep}(R') = \max\big(\text{Dep}(R_i^{d_i}), \text{Dep}(R_j^{d_j})\big) + 1, \tag{4}$$

where $\text{Dep}(R_i^{d_i}) = d_i$ and $\text{Dep}(R_j^{d_j}) = d_j$. This recursive definition naturally reflects the hierarchical buildup of reward components as evolution progresses. Intuitively, deeper rewards encode more subcomponents and thus capture increasingly sophisticated shaping strategies. Compared to iteration indices, which merely record creation order, Dep provides a principled and fine-grained measure of complexity. As we will later show, this enables us to analyze how reward complexity correlates with performance across different environments.

## 3.2 REWARDS EVOLUTION

With the initial reward population organized in the pool and their structural complexity formally defined, the next step is to improve these candidates through iterative refinement. To this end, FORGE employs an evolutionary process that selectively combines high-performing rewards and explores new ones, progressively increasing both the diversity and the effectiveness of the population.

The classical genetic algorithm (GA) (Holland, 1992) represents candidate solutions (or "chromosomes") as fixed-length bit strings, where each bit encodes a specific feature. New candidates are then generated through crossover and mutation between two selected parents. While effective in such discrete settings, this formulation does not directly transfer to reward functions for two main reasons. First, reward functions vary in length and structure, making it difficult to impose a homogeneous encoding. Second, naive recombination of code fragments is inefficient and severely limits exploration. For instance, one might attempt to represent each reward component from subsection 3.1 as a binary feature—"1" for active, "0" for inactive. This rigid representation assigns equal weight to all components and leads to an intractable search space when extended to real-valued weightings.

To address these limitations, we generalize the evolutionary process by incorporating LLM inference. Rather than operating on fixed encodings, our framework leverages the reasoning capability of LLMs to guide both the selection and crossover of reward functions. The following sections detail these two mechanisms.

**Rewards Selection.** We maintain a function pool $\mathcal{S}$ that extends the initial reward population $\mathbb{R}$ introduced in subsection 3.1. While $\mathbb{R}$ contains only the planner-sampled rewards (depth $= 0$), $\mathcal{S}$ dynamically grows throughout evolution by incorporating newly generated candidates and discarding weaker ones. Formally, at iteration $t$, the pool is represented as

$$\mathcal{S} = \{(R_1, J^*_{R_1}), (R_1, J^*_{R_1}), \ldots, (R_n, J^*_{R_n})\} \tag{5}$$

where $n$ is the number of functions currently stored and $J_{R_i}$ denotes $J(\pi_{R_i}, \bar{R})$ for brevity. The asterisked $J^*$ implies the best return over all evaluation episodes.

In contrast to traditional GA, where a fixed fitness function determines selection, we use the scores $J^*_{R_i}$ not only for evaluation but also as sampling weights. Since each crossover requires two parents, we define a categorical distribution over all pairs $(R_i, R_j)$ in $\mathcal{S}$:

$$\mathbb{P}((R_i, R_j)) = \begin{cases} \frac{J^*_{R_i} + J^*_{R_j}}{W}, & \text{if } W > 0, \\ \frac{1}{K}, & \text{if } W = 0 \quad \text{(uniform sampling)}. \end{cases} \tag{6}$$

where $K$ is the total number of functions in the pool and $W = \sum_k J^*_{R_k}$ is the sum of all scores. To maintain bounded size, the lowest-scoring functions are pruned from $\mathcal{S}$ after each iteration.

This probabilistic selection scheme introduces stochasticity into the process, enabling both exploitation of high-performing rewards and exploration of new combinations, thereby supporting more effective evolution of the reward population.

**Rewards Crossover.** Once parent rewards are sampled from the pool $\mathcal{S}$, new candidates are generated through crossover. Unlike classical GA, which recombines fixed encodings, we adopt a high-level approach that leverages the reasoning and coding ability of LLMs. Given two parent rewards $R_i$ and $R_j$, along with their scores, a coder agent synthesizes an offspring $R'$ by interpreting their semantic content:

$$R' \sim \mathbb{P}_\theta(\cdot | R_i, R_j, J^*_{R_i}, J^*_{R_j}). \tag{7}$$

To ensure compatibility across environments, we standardize the crossover operation through fixed function interfaces, which specify argument types and return values. This allows the coder agent to focus solely on implementing the reward logic, without concerns about environment-specific details. Moreover, the design is memory-less: the current environment states (e.g., observations and actions) are passed directly as arguments, eliminating dependence on long historical context.

The evolution proceeds iteratively. At iteration $t$, the coder samples $K$ pairs of rewards from $\mathcal{S}_t$, generates new candidates $R_1^{(t)}, R_2^{(t)}, \ldots, R^{(t)}K$, and updates the pool as

$$\mathcal{S}_{t+1} = \mathcal{S}_t \cup \{R_1^{(t)}, R_2^{(t)}, \ldots, R_K^{(t)}\}. \tag{8}$$

The process continues until a predefined number of iterations is reached. Because each generation relies only on local context (the two parents and their scores), the risk of hallucination or excessive token usage is minimized. In addition, invalid functions can be safely discarded without disrupting the population, avoiding the need for explicit error-handling mechanisms.

## 4 EXPERIMENTS

### 4.1 ENVIRONMENTS

We evaluate FORGE across four representative environments: three games—Tetris (Weichart & Hartl, 2024), Snake (Grant, 2023), and Flappy Bird (Kubovčík, 2024)—and one simulated robotics task, the humanoid robot in the Gymnasium (Towers et al., 2024) MuJoCo (Todorov et al., 2012) suite. These environments are chosen to cover both discrete and continuous control settings, thereby testing the generality of our approach. Since reward functions are generated with an LLM, we emphasize the

Table 1: Summary of the observation space for training RL policy and information received by the reward functions in addition to the observations.

| Environment | RL Policy Observation | Additional Reward Arguments |
|---|---|---|
| Tetris | $\{0, 1\}^{200}$, flattened game grid | past observation and tetromino position |
| Snake | $\mathbb{R}^5$, snake and food positions | past observation and game states |
| Flappy Bird | $\mathbb{R}^{12}$, player and pipes positions | past observation and game states |
| Humanoid | $\mathbb{R}^{348}$, humanoid kinematics | past observation, action and external forces |

distinction between the observation used as input to the RL policy and the input to the LLM-based reward function, which includes additional information necessary to construct reward signals.

Table 1 summarizes the input spaces of all environments. For evaluation, we report the mean cumulative extrinsic reward over $N$ evaluation episodes. In the gaming environments, the extrinsic rewards correspond directly to the game objectives: the number of lines cleared in Tetris, the snake length in Snake, and the number of pipes passed in Flappy Bird. In the humanoid environment, the extrinsic reward is a composite measure consisting of the alive reward, forward-movement reward, control cost, and contact penalty. We include a **video demonstration of FORGE** across four environments in the supplementary material.

### 4.2 BASELINES

**Eureka.** Eureka (Ma et al., 2024a) is a self-improving framework to encourage LLM to generate executable reward functions and iteratively refine their design. In each iteration, Eureka selects the best-performing reward from the previous round, produces a textual reflection, and queries the LLM to generate $K$ additional reward functions. For fair comparison, we run Eureka for 10 iterations with $K = 16$, and report the highest extrinsic reward achieved as its final score.

**REvolve.** REvolve (Hazra et al., 2025) incorporates human feedback to quantitatively evaluate LLM-generated reward functions. Instead of greedily generating new functions, REvolve employs a strategy that selectively combines or mutates existing functions via the LLM. To ensure comparability, we replace the human feedback module with the extrinsic reward of the target environment, and run 10 iterations where 16 (i.e., $K = 16$) reward functions are refined in each iteration.

**General Agentic Frameworks.** These are the general-purpose agents that have access to resources online. We include two state-of-the-art frameworks: MetaGPT (Hong et al., 2024) and ChatGPT Agent (OpenAI, 2025a). With zero-shot prompting, we ask the frameworks to implement a dense reward function for each environment.

**Context-aware LLMs.** As recent LLMs are getting larger, they can process much longer prompts. This capability enables single LLMs to zero-shot generate reward functions, and iteratively improve the reward functions given feedbacks. We include Claude 4 (Anthropic, 2025), Grok 4 (xAI, 2025), GPT-5 (OpenAI, 2025b) and o3 (OpenAI, 2025c), and run 10 iterations for each model.

**Native.** Environment-provided rewards without LLM shaping: sparse, event-driven signals in games (e.g., lines in *Tetris*, apples in *Snake*, pipes in *Flappy Bird*) with terminal penalties, and dense composite rewards in *Humanoid* (alive bonus, velocity, control cost, contact penalty).

### 4.3 TRAINING DETAILS

**Policy Training.** All RL policies are trained using the Stable-Baselines3 (Raffin et al., 2021) implementation of Proximal Policy Optimization (PPO) (Schulman et al., 2017), with a fixed set of hyperparameters across all environments. While alternative algorithms or tuned hyperparameters may yield stronger performance, we adopt this unified setting to ensure consistency, fairness, and reproducibility in our comparisons.

---

[1]General agentic frameworks (e.g., ChatGPT Agent, MetaGPT) do not provide reward function self-improvement mechanisms, and thus only report Initialized results.

Table 2: **Performance comparisons** across four environments. "Zero-shot" refers to policies trained with LLM-generated rewards without refinement, while "Evolved" denotes performance after iterative evolutionary updates. Scores are reported as the mean extrinsic rewards obtained by trained policies. Forge consistently outperforms all baselines, achieving the best results in the evolved settings.[1]

| Environment | Tetris | | Snake | | Flappy Bird | | Humanoid | |
|---|---|---|---|---|---|---|---|---|
| mode | Zero-shot | Evolved | Zero-shot | Evolved | Zero-shot | Evolved | Zero-shot | Evolved |
| Native | 4.2 | - | **17.0** | - | 36.8 | - | 696.7 | - |
| Claude Sonnet 4 | 0.0 | 2.6 | 12.0 | 15.4 | 15.8 | 59.6 | 642.6 | 775.1 |
| Grok 4 | 0.0 | 7.6 | 10.2 | 15.6 | 51.6 | 7.8 | 621.2 | 671.5 |
| o3 | 0.0 | 3.8 | 14.0 | 14.8 | 9.2 | 4.4 | 680.7 | 699.5 |
| GPT-5 | 0.0 | 5.0 | 14.8 | 17.4 | 10.8 | 90.8 | 645.4 | 716.0 |
| ChatGPT Agent | 0.0 | - | 16.0 | - | 68.0 | - | 579.0 | - |
| MetaGPT | 0.0 | - | 14.0 | - | 5.2 | - | 651.2 | - |
| Eureka | 0.0 | 6.8 | 10.2 | 19.2 | 49.0 | 114.8 | 105.8 | 756.7 |
| REvolve | 1.2 | 9.4 | 14.0 | 17.2 | 32.4 | 129.2 | 686.7 | 880.7 |
| Forge (Ours) | **8.8** | **11.2** | 6.0 | **19.8** | **86.0** | **254.8** | **815.2** | **1048.0** |

**Self-Improvement.** For frameworks with self-improving mechanisms (AgentRF, Eureka, REvolve, and Context-aware LLMs), we run each method for 10 iterations. In population-based sampling, $K = 16$ candidate reward functions are generated at each iteration, from which the selection or evolution process is applied.

**Foundation Models.** To ensure fair comparison, we reproduce all baseline methods and run them under the same setting, using the CLAUDE-SONNET-4-20250514 variant of Claude Sonnet 4 (Anthropic, 2025) as the foundation model for both our framework and the baselines.

## 4.4 RESULTS

**Performance Comparisons.** We report the main experimental results across four environments in Table 2. FORGE achieves the strongest performance overall, outperforming both general-purpose LLMs (Claude Sonnet 4, Grok 4, o3, GPT-5) and specialized frameworks (Eureka (Ma et al., 2024a), REvolve (Hazra et al., 2025)) in refining rewards. In the gaming environments, FORGE demonstrates significant improvements over the zero-shot baselines, particularly in *Flappy Bird*, where iterative refinement yields more than a $3\times$ increase compared to the strongest baseline. In the humanoid task, FORGE not only surpasses the native environment reward but also consistently outperforms self-improving baselines such as Eureka and REvolve. These results highlight Forge's ability to generate high-quality reward functions at initialization and further enhance them through iterative refinement.

**FORGE steadily refines reward quality.** While Table 2 reports the final evolved rewards across environments, Figure 2 illustrates the intermediate refinement process. FORGE consistently produces higher-quality reward populations than baseline methods and maintains stable improvements over iterations. A key observation from Figure 2 is that population-based approaches, such as REvolve and Eureka, do not, on average, surpass naive LLM-based methods by a large margin, except in isolated cases such as Tetris. Although these methods can eventually discover reward functions that maximize returns (as shown in Table 2), the process is costly and unstable, and in many cases context-aware LLMs perform at a comparable average level. In contrast, FORGE exhibits stronger adaptability and demonstrates consistent refinement, achieving at least the average level of all tested baselines while steadily improving reward quality across iterations.

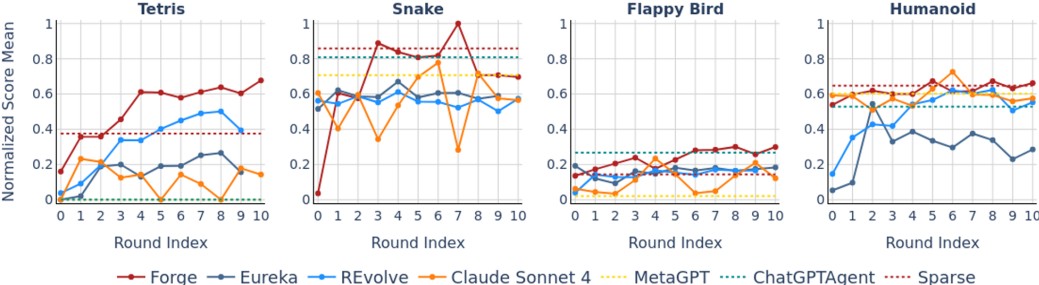

Figure 2: Means of the reward function scores over the evolution process. Each data point represents the mean of all reward function scores at the corresponding round index. The scores are normalized across all evaluated methods for the corresponding environment. For clarity, we show only the data for Claude Sonnet 4 from all context-aware LLMs.

## 4.5 DISCUSSION

**Ablation Study.** FORGE is designed as a compact framework without clearly separated modules, which makes ablation less straightforward than in modular systems. To better understand the contribution of its core components, we perform two ablation studies by selectively removing key design choices. Specifically, we analyze the effect of removing the *selective evolution* and *rewards initialization planning* modules, which correspond to disabling the Engineer agent and the Planner agent, respectively. Table 3 reports the ablation study results across four environments. We observe that removing either component leads to noticeable drops in performance. Without selective evolution, FORGE struggles to maintain high scores in Snake and Humanoid, highlighting the importance of guided refinement. Without rewards initialization planning, performance degrades substantially in Flappy Bird and Humanoid, showing that sampling diverse reward components plays a crucial role in stabilizing long-horizon optimization. The full FORGE framework, with both components enabled, consistently achieves the best results across all environments, confirming that each module is essential for maximizing performance.

Table 3: Ablation study of Forge across four environments. We analyze the contribution of two core modules, selective evolution (Select. Evolve) and rewards initialization planning (Rewards Init.), by selectively removing each component. Results show that removing either module leads to noticeable drops in performance. The details of the two modules are elaborated in the main text.

| Ablation Study | Modules | | Environments | | | |
|---|---|---|---|---|---|---|
| | Select. Evolve | Rewards Init. | Tetris | Snake | Flappy Bird | Humanoid |
| w/o Select. Evolve | | ✓ | 10.6 | 13.8 | 151.4 | 872.7 |
| w/o Rewards Init. | ✓ | | 10.0 | 19.6 | 126.8 | 635.1 |
| Forge | ✓ | ✓ | **11.2** | **19.8** | **254.8** | **1048.0** |

**Depth analysis of reward functions.** As defined in Equation 4, we use depth to measure the structural complexity of generated reward functions. Figure 3 presents the relationship between depth and performance. In the gaming environments (Figure 3a), we observe that the best-performing reward functions consistently occur at depths around 3, despite the evolutionary process continuing to generate deeper functions. This suggests that simple yet well-structured rewards are sufficient for these discrete control tasks, and excessive complexity does not yield additional benefits. In contrast, for the humanoid environment (Figure 3b), the optimal reward functions emerge at depth 7, indicating that more sophisticated compositions are necessary to capture the continuous and high-dimensional dynamics. Moreover, the results highlight that FORGE converges stably across diverse levels of complexity, whereas context-aware LLMs (see Figure 2) exhibit fluctuating performance, particularly in Tetris and Snake. Finally, the depth metric itself provides a useful signal for efficient search: when performance stagnates as depth increases, the process can be terminated early, avoiding unnecessary exploration of overly complex reward functions.

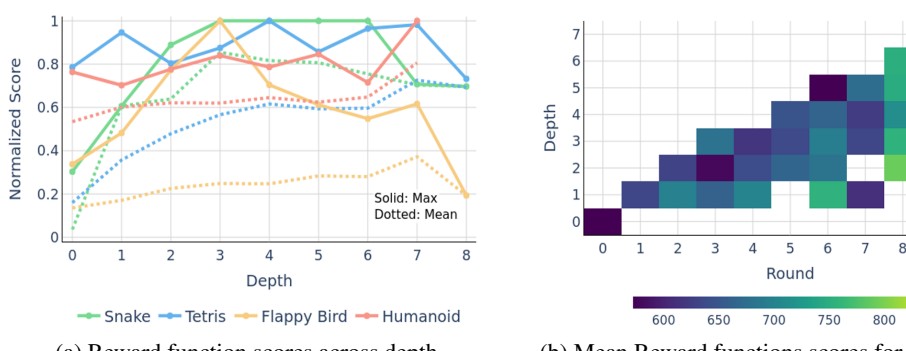

(a) Reward function scores across depth.  (b) Mean Reward functions scores for Humanoid.

Figure 3: Depth analysis of reward functions. (a) Normalized scores across depths for four environments, showing that optimal depths differ between games and humanoid control. (b) Distribution of mean reward scores by depth and round in Humanoid.

**Performance gain does not come at the expense of computation cost.** Recent works on multi-agents collaborations suggest the trend of increasing token consumption with performance gain (Yuan et al., 2025; Qian et al., 2025). As shown in Figure 7, our approach consumes more tokens in two out of four tested environments but achieves consistent gains across all environments. This result is primarily attributed to the design of the evolution strategy: the LLM is constrained to modify only a small portion of the population on each iteration, and the context of generating a new reward is limited to two sampled parent functions. This design also excludes the need for error-checking mechanisms, as unsuccessful reward functions are simply discarded without affecting the population. In extreme cases where the LLM generates mostly non-executable functions, our approach falls back to repetitively sampling responses from the LLM without incurring additional cost, since all stored functions have chances to be drawn.

**Token Efficiency.** Prior work on multi-agent collaboration has suggested that performance gains often come with increased token consumption (Yuan et al., 2025; Qian et al., 2025). Nevertheless, FORGE attains significant improvements with only a modest increase in token consumption. (Analysis is detailed in Appendix A). This efficiency arises from the evolution strategy: at each iteration, the LLM modifies only a small subset of the population, and the context for generating a new reward is limited to two sampled parent functions. As a result, FORGE requires no additional error-checking—invalid functions are simply discarded without affecting the population—and can fall back to resampling in extreme cases where most generations are invalid, without incurring extra cost.

## 5 CONCLUSION

In this paper, we introduced FORGE, a multi-agent framework for automated reward evolution that integrates planner-based initialization, evolutionary refinement, and complexity modeling. Experiments across games and robotics show consistent gains over strong baselines, including up to **38.5%** improvement on the Humanoid benchmark, while maintaining token efficiency. Beyond empirical performance, FORGE highlights how LLMs can be "forged" through feedback: by grounding reward evolution in numerical returns and structural depth, the framework exploits reasoning abilities that go beyond natural language and into domains where LLMs have traditionally struggled. This opens promising directions for reward design in real-world robotics, where scalable, interpretable, and feedback-driven reasoning will be critical for embodied AI.

**Limitations and future work.** The main limitation of FORGE is that it has only been evaluated in simulated environments. Demonstrating success in real-world robotics is essential to establish the practical effectiveness of automated reward shaping, as physical systems introduce challenges such as sensor noise, delayed feedback, and strict safety constraints. In the longer term, scaling FORGE to real-world deployment will likely require LLMs not only to generate and refine reward functions but also to adapt their own reasoning through fine-tuning on embodied feedback. This suggests a future trajectory where reward shaping and LLM adaptation co-evolve, ultimately bridging the gap between simulation-driven design and robust real-world intelligence.

## LARGE LANGUAGE MODELS USAGE STATEMENT

Large Language Models (LLMs) were used solely to assist with the linguistic polishing of this manuscript, such as improving grammar, clarity, and readability. All conceptual contributions, technical methods, experimental designs, and analyses were developed entirely by the authors without the use of LLMs.

## ETHICS STATEMENT

This work does not involve human subjects, personally identifiable data, or sensitive information. All experiments are conducted in simulated environments (video games and robotics benchmarks) and therefore raise no direct ethical or privacy concerns. The proposed methods aim to improve the efficiency and interpretability of reinforcement learning without foreseeable harmful applications. We have carefully adhered to the ICLR Code of Ethics throughout the research and writing process.

## REPRODUCIBILITY STATEMENT

We have made significant efforts to ensure the reproducibility of our results. A detailed description of the algorithm, including mathematical formulations and pseudo-code, is provided in the main paper and appendix. The prompts used for LLM-based reward generation are included in the supplementary materials. The full source code, together with configuration files for all experiments, will be released upon publication of the final version of the paper.

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

# A ADDITIONAL ANALYSIS

In this section, we provide analyses of (i) how FORGE obtains the optimal reward through a synthesis process, (ii) a comparison between the optimal reward generated by FORGE and other methods, and (iii) complete mean reward function scores and token usage omitted in subsection 4.4.

## A.1 REWARD SYNTHESIS

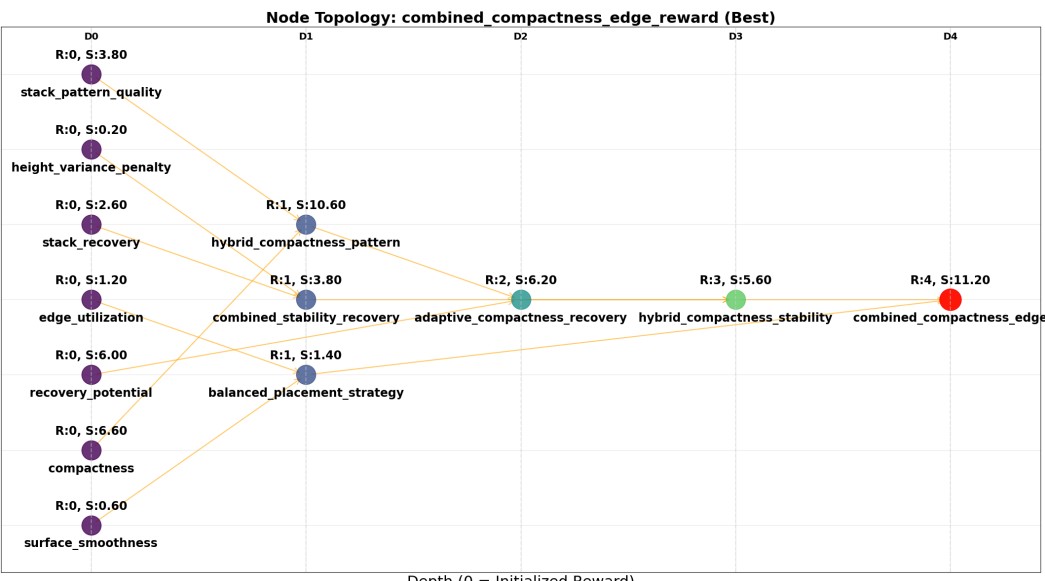

Figure 4: Node topology for Tetris. Each node is a reward function labeled by the LLM generated name, round index (R) and score (S). Each edge represents a parent-child relation. Due to the crossover operation requires exactly two parents to create a new child, every node is connected by two edges.

In Figure 4, we show the evolution process that synthesizes the optimal reward function for Tetris. The reward functions shown are plotted with increasing depth, representing an increasing degree of complexity but not necessarily better performance. Note that the parent reward functions and their scores are presented to the agents to create new rewards, so the evolution process is *Markovian*. For illustrative purpose, only the topology for the best rewards of Tetris and Humanoid are shown. We list a sub-path in the following table to demonstrate the synthesis of best reward for Tetris, with a summary for each generated reward function.

| **Example Sub-Path in Tetris Topology** | |
|---|---|
| **Edge Utilization Reward** | Small reward for effectively using board edges for piece placement. Encourages strategic use of wall kicks and edge positioning to maximize placement efficiency. |
| **Surface Smoothness Reward** | Reward for maintaining relatively flat surfaces with minimal height differences between adjacent columns. Promotes board states that provide flexible placement options for future pieces. |
| **Balanced Placement Strategy Reward** | Combines edge utilization with surface smoothness to encourage strategic piece placement. Primarily rewards edge usage for stability while penalizing excessive surface roughness when the agent is performing well, promoting both immediate placement efficiency and long-term board management. |
| **Combined Compactness Edge Reward** | Combines the successful compactness-stability foundation with strategic edge utilization. Uses compactness and adaptive stability as the primary drivers for consistent line clearing performance. |

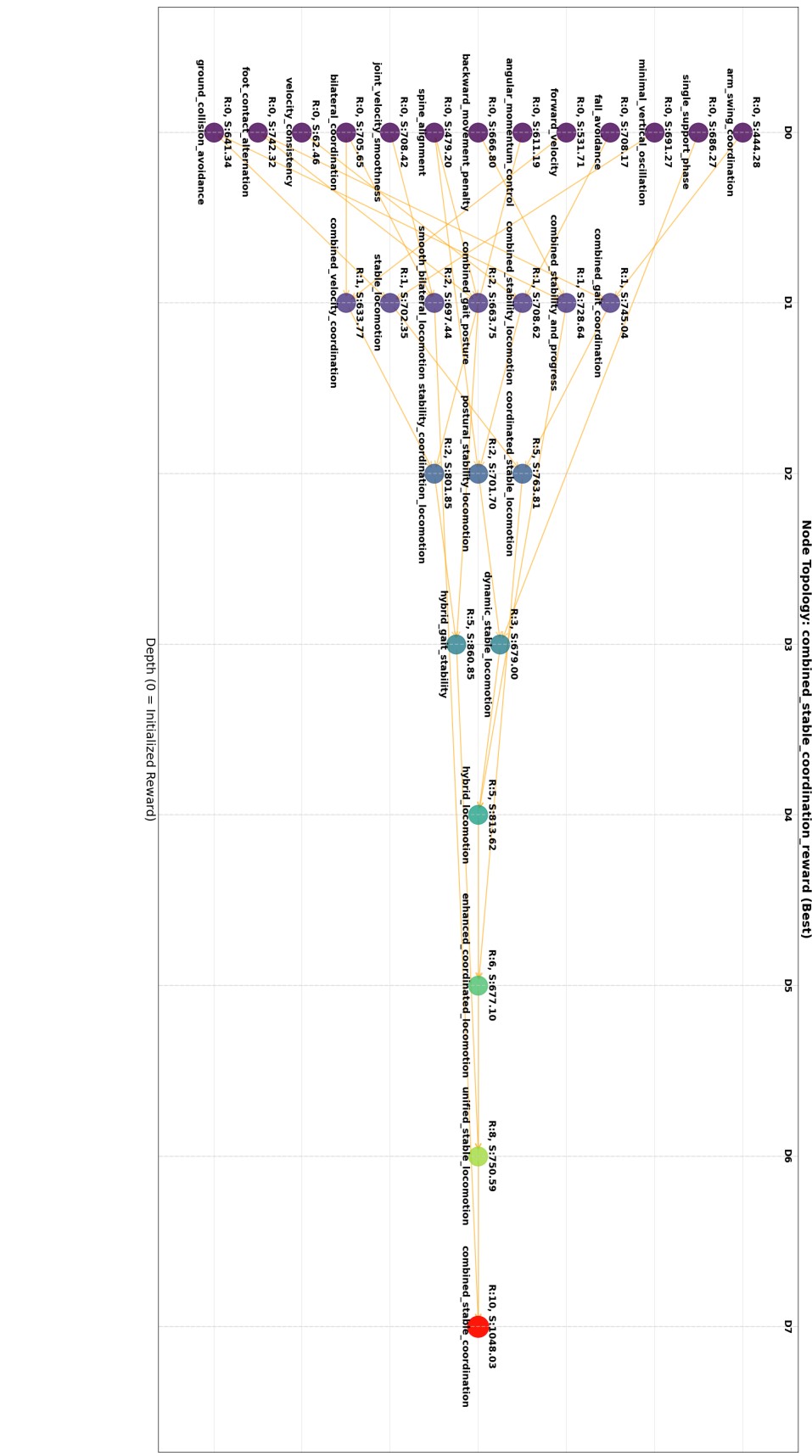

Figure 5: Node topology for Humanoid.

## A.2 REWARDS COMPARISON

This section details the difference between the optimal rewards generated for FORGE, REOLVE and EUREKA using pseudo code. For illustrative purposes, only the summarization for the humanoid environment is included. As shown in the following pseudo code summarization, a major difference between FORGE and the other methods is the inclusion of coordination components that facilitate human-like moving pattern. On the other hand, there are components common to all three method (e.g. angular momentum penalty and forward moving reward), which semantically align with the Humanoid objective, i.e. moving forward and not falling. We also present the complete source code of the Humanoid optimal reward obtained by FORGE.

---

**Algorithm 1** FORGE OPTIMAL REWARD

---

1: **function** FORGEREWARD
2:     **# 1. Collision penalty**
3:     **for** each $non\_foot\_part$ **do**
4:         **if** any(contact) **then**
5:             $reward \leftarrow -3.0$
6:         **end if**
7:     **end for**
8:
9:     **# 2. Penalize large angular momentum**
10:     $reward \leftarrow -0.015 \cdot \left( \sum |\omega_{torso}| + \sum |\omega_{abdomen}| \right)$
11:
12:     **# 3. reward bilateral coordination patterns**
13:     $reward \leftarrow 1.0 - \dfrac{|x_{right\_hip} + x_{left\_hip}|}{1.0}$           ▷ Legs Coordination
14:     $reward \leftarrow max(0,\ 2 - \dfrac{|x_{right\_arm} + x_{left\_hip}|}{1.0} - \dfrac{|x_{left\_arm} + x_{right\_hip}|}{1.0})$
15:               ▷ Synchronous arms and legs
16:     $reward \leftarrow 1.0 - \dfrac{||v_{right\_hip}| - |v_{left\_hip}||}{5.0}$           ▷ Synchronous hips
17:
18:     **# 4. Joint velocity smoothness over time**
19:     **for** each joint $v_t, v_{t-1}$ **do**
20:         $reward \leftarrow -0.006 \cdot \sum (v_t - v_{t-1})^2$
21:     **end for**
22:
23:     **# 5. Alternating foot contact reward**
24:     **if** $right\_foot\_contact\ or\ left\_foot\_contact$ **then**
25:         $reward\ += 0.25$
26:     **end if**
27:
28:     **return** $reward$
29: **end function**

---

---

**Algorithm 2** REVOLVE OPTIMAL REWARD

---

1: **function** REVOLVEREWARD
2:  $reward = 0$
3:  $reward \leftarrow \begin{cases} healthy\_bonus & \text{if } is\_healthy \\ 0.0 & \text{otherwise} \end{cases}$
4:
5:  **# 1. Temperature-scaled action smoothness with velocity dependence**
6:  $velocity\_scale \leftarrow 1.0 + 0.4 \cdot |forward\_velocity|$
7:  $reward \leftarrow -temp \cdot velocity\_scale \cdot \left(1 - e^{-action\_diff/temp}\right)$
8:
9:  **# 2. Graduated exponential contact penalties**
10:  $reward \leftarrow -temp \cdot \left(1 - e^{-\sum contact\_forces}\right)$
11:
12:  **# 3. orientation stability**
13:  $reward \leftarrow -temp \cdot (torso\_orientation + angular\_velocity\_penalty)$
14:
15:  **# 4. Forward reward with tanh transform**
16:  $reward \leftarrow temp \cdot \tanh(\frac{v_{torso}}{temp})$
17:
18:  **# 5. efficiency and consistency bonuses**
19:  $reward \leftarrow$ REWARDFORWARDLATERALMOVEMENT
20:  $reward \leftarrow$ REWARDGOODPOSTUREWHILEMOVING
21:
22:  **return** $reward$
23: **end function**

---

**Algorithm 3** EUREKA Optimal Reward

---

1: **function** EUREKAREWARD
2:  **// 1. Weighted, smooth contact penalties**
3:  **for** each $non\_foot\_part$ **do**
4:    $reward \leftarrow -weight[body\_part] \cdot \tanh(temp \cdot |F_{body\_part}|^2)$
5:  **end for**
6:
7:  **// 2. Angular velocity based stability**
8:  $\omega_{excess} \leftarrow \max\left(0, |\omega_{torso}| + |\omega_{abdomen}| - threshold\right)$
9:  $reward \leftarrow -temp \cdot \sum(\omega_{excess}^2)$
10:
11:  **// 3. Forward locomotion**
12:  $reward \leftarrow \tanh(temp \cdot v_{torso})$
13:
14:  **// 4. Upright posture**
15:  $reward \leftarrow temp \cdot \exp\left(-(z_{torso} - 1)^2\right)$
16:
17:  **return** $reward$
18: **end function**

---

**FORGE: Humanoid Optimal Reward**

```
def combined_stable_coordination_reward(score_info: dict, action: np.
    ndarray, prev_action: np.ndarray, x_coord: float, y_coord: float,
    prev_x_coord: float, prev_y_coord: float, distance_from_origin:
    float, prev_distance_from_origin: float, healthy_z_range: Tuple,
    obs: np.ndarray, prev_obs: np.ndarray, cfrc_ext: dict,
    prev_cfrc_ext: dict, terminated: bool, truncated: bool) -> float:
    """Combined stable coordination reward that merges smooth
        bilateral locomotion with robust stability control.
```

```
This reward function combines the bilateral coordination patterns
    and joint velocity smoothness
from the smooth_bilateral_locomotion_reward with the strong
    collision avoidance and angular
stability control from the unified_stable_locomotion_reward. It
    promotes natural human-like
gait through coordinated limb movement while maintaining robust
    fall prevention and collision
penalties for consistent training progress."""
# Base reward from environment
base_reward = (score_info.get('healthy_reward', 0.0) +
                score_info.get('forward_reward', 0.0) +
                score_info.get('ctrl_cost', 0.0) +
                score_info.get('contact_cost', 0.0))

# Early termination handling with stability penalty
if terminated:
    return base_reward - 8.0  # Strong termination penalty

# Get torso height for health check
torso_height = obs[0]
is_healthy = healthy_z_range[0] <= torso_height <= healthy_z_range
    [1]

if not is_healthy:
    return base_reward - 5.0

# Component 1: Joint Velocity Smoothness (from
    smooth_bilateral_locomotion_reward)
current_joint_velocities = obs[25:45]
prev_joint_velocities = prev_obs[25:45]
velocity_changes = current_joint_velocities -
    prev_joint_velocities
velocity_change_magnitudes = np.abs(velocity_changes)
smoothness_penalty = np.sum(velocity_change_magnitudes ** 2)
smoothness_reward = -0.006 * smoothness_penalty  # Slightly
    reduced weight

# Component 2: Bilateral Coordination (from
    smooth_bilateral_locomotion_reward)
coordination_reward = 0.0

# Extract joint angles and velocities
right_hip_x, right_hip_z, right_hip_y, right_knee = obs[8:12]
left_hip_x, left_hip_z, left_hip_y, left_knee = obs[12:16]
right_arm_1, right_arm_2 = obs[16:18]
left_arm_1, left_arm_2 = obs[19:21]

right_hip_x_vel, right_hip_z_vel, right_hip_y_vel, right_knee_vel
    = obs[31:35]
left_hip_x_vel, left_hip_z_vel, left_hip_y_vel, left_knee_vel =
    obs[35:39]

# Anti-phase leg movement (natural gait)
leg_phase_diff = abs(right_hip_x + left_hip_x)
leg_coordination = max(0, 1.0 - leg_phase_diff / 1.0)
coordination_reward += leg_coordination * 0.2

# Arm-leg coordination (arms opposite to legs)
right_arm_left_leg_sync = 1.0 - abs(right_arm_1 - left_hip_x) /
    2.0
left_arm_right_leg_sync = 1.0 - abs(left_arm_1 - right_hip_x) /
    2.0
```

```
         arm_leg_coordination = max(0, (right_arm_left_leg_sync +
             left_arm_right_leg_sync) / 2.0)
         coordination_reward += arm_leg_coordination * 0.1

         # Velocity symmetry between limbs
         hip_vel_coordination = max(0, 1.0 - abs(abs(right_hip_x_vel) - abs
             (left_hip_x_vel)) / 5.0)
         knee_vel_coordination = max(0, 1.0 - abs(abs(right_knee_vel) - abs
             (left_knee_vel)) / 5.0)
         velocity_coordination = (hip_vel_coordination +
             knee_vel_coordination) / 2.0
         coordination_reward += velocity_coordination * 0.1

         # Component 3: Foot Contact Patterns (from
             unified_stable_locomotion_reward)
         right_foot_force = np.linalg.norm(cfrc_ext['right_foot'][:3])
         left_foot_force = np.linalg.norm(cfrc_ext['left_foot'][:3])

         right_contact = right_foot_force > 0.3
         left_contact = left_foot_force > 0.3

         # Reward single foot contact (natural walking)
         if (right_contact and not left_contact) or (not right_contact and
             left_contact):
             coordination_reward += 0.25
         elif right_contact == left_contact:
             coordination_reward -= 0.05

         # Contact force symmetry when both feet in contact
         if right_foot_force + left_foot_force > 0.1:
             force_asymmetry = abs(right_foot_force - left_foot_force) / (
                 right_foot_force + left_foot_force + 1e-6)
             force_symmetry = max(0, 1.0 - force_asymmetry)
             coordination_reward += force_symmetry * 0.08

         # Component 4: Stability Control (from
             unified_stable_locomotion_reward)
         stability_reward = 0.0

         # Strong collision avoidance for non-foot body parts
         contact_threshold = 1e-6
         collision_penalty = -3.0

         non_foot_body_parts = [
             'torso', 'lwaist', 'pelvis',
             'right_thigh', 'right_shin', 'left_thigh', 'left_shin',
             'right_upper_arm', 'right_lower_arm', 'left_upper_arm', '
                 left_lower_arm'
         ]

         for body_part in non_foot_body_parts:
             if body_part in cfrc_ext:
                 contact_force_magnitude = np.linalg.norm(cfrc_ext[
                     body_part])
                 if contact_force_magnitude > contact_threshold:
                     stability_reward += collision_penalty

         # Angular stability control
         torso_angular_vel = obs[25:28]
         abdomen_angular_vel = obs[28:31]

         torso_momentum = np.sum(np.abs(torso_angular_vel))
         abdomen_momentum = np.sum(np.abs(abdomen_angular_vel))
```

```
        stability_reward -= 0.015 * (torso_momentum + 0.8 *
            abdomen_momentum)

        # Postural control
        torso_w, torso_x, torso_y, torso_z = obs[1:5]
        abdomen_x, abdomen_y = obs[7], obs[6]

        # Torso pitch control
        torso_pitch = 2 * (torso_w * torso_y - torso_z * torso_x)
        if abs(torso_pitch) > 0.3:
            stability_reward -= 0.2

        # Spine stability
        if abs(abdomen_x) > 0.15 or abs(abdomen_y) > 0.15:
            stability_reward -= 0.1

        # Backward movement penalty
        x_velocity = x_coord - prev_x_coord
        if x_velocity < 0:
            stability_reward -= 2.0 * abs(x_velocity)

        # Component 5: Forward velocity bonus
        forward_velocity = obs[22]
        velocity_bonus = 0.1 * min(forward_velocity, 2.0) if
            forward_velocity > 0 else 0

        # Combine all components with balanced weighting
        total_reward = base_reward + smoothness_reward +
            coordination_reward + stability_reward + velocity_bonus

        return total_reward
```

## A.3   REWARD FUNCTION SCORES AND TOKEN USAGE.

The mean reward functions scores for the environments omitted in subsection 4.5 are shown in Figure 6.

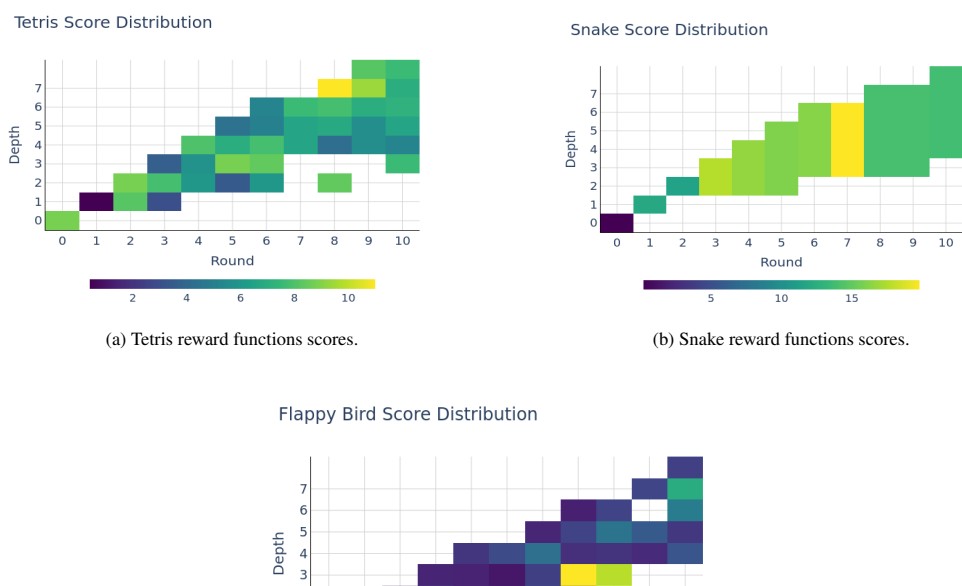

(a) Tetris reward functions scores.

(b) Snake reward functions scores.

(c) Flappy Bird reward functions scores.

Figure 6: Reward functions scores plotted across depth and round.

It is evident that environments with more complex dynamics have the optimal reward functions occurred deeper in the evolution process. A notable observation is the Flappy Bird envrionment, where the optimal reward appears as an outlier that performs surprisingly better than all other rewards. This sharp local maxima within the solution space might suggests critical reward components that dramatically improve the policy performance.

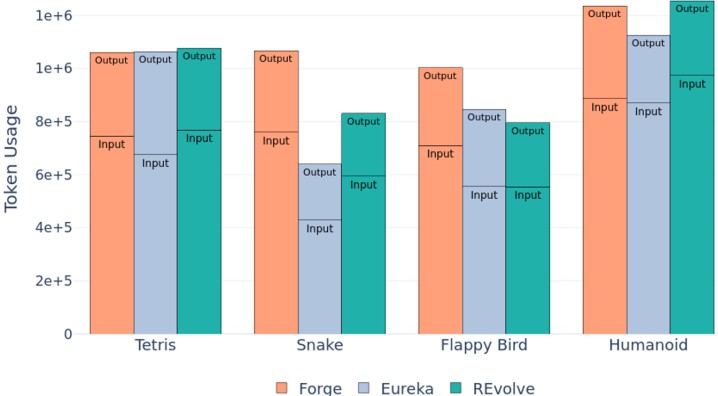

Figure 7: Token usage comparison across four environments. Bars are split into input and output tokens for each method. FORGE consumes more tokens in only two environments, yet achieves consistent performance gains across all cases, demonstrating its token efficiency relative to Eureka and REvolve.

# B   FORGE DETAILS

In the following sections we present the implementation detail of FORGE with i) the detailed algorithm, ii) environments abstraction, and iii) reward function interfaces.

## B.1   FORGE ALGORITHM

---

**Algorithm 4** Forge

---

**Require:** Environment definition $env$, reward function interface $I$

**Hyperparameters:** number of initial rewards sampling $N$, total evolution rounds $T$, number of new rewards $K$

**Ensure:** Evolved reward function population

1: **Initialize** AI agents:
2:     reward planner P                                        ▷ Sample reward component ideas
3:     reward engineer E                                      ▷ Implements functions from specifications
4: # Initialize Reward Population
5: $\mathcal{S} := \{\}$                                                              ▷ Empty reward function pool
6: $reward\ specs := \{P(env)_i\}_{i=1}^N$                         ▷ Planner brainstorms reward components
7: **for** $spec \in reward\ spec$ **do**
8:       $R := P(spec, I)$                                              ▷ Sample initial rewards
9:       $J_R^* = Train(\pi, \bar{R})$                       ▷ Score of $R$ is the maximum native reward $\bar{R}$
10:       $\mathcal{S} \leftarrow \mathcal{S} \bigcup (R, J_R)$
11: **end for**
12: # Iterative training and evolution
13: **for** $T$ rounds **do**
14:       **for** $t =$ start_round to $T$ **do**
15:             $\{(R_i, R_j)\}^K \sim Categorical(\mathcal{S},\ p(J_{R_i}^*, J_{R_j}^*))$          ▷ Sample reward function pairs
16:             **for** $(R_i, R_j) \in \{(R_i, R_j)\}^K$ **do**
17:                   $R' = E((R_i, R_j))$                                        ▷ Obtain child reward $R'$
18:                   $J_{R'}^* = Train(\pi, \bar{R})$
19:                   $\mathcal{S} \leftarrow \mathcal{S} \bigcup (R', J_{R'}^*)$
20:             **end for**
21:       **end for**
22: **end for**
23: $R^* := \underset{J^*}{argmax}\ \mathcal{S}$                    ▷ Optimal reward is the reward function with the highest score
24: **return** $\mathcal{S}$

---

## B.2   ENVIRONMENTS ABSTRACTION

FORGE abstract each environment as simple textual description. This abstraction facilitates exploration in initializing reward population, as the potential rewards are not constrained by the extra context required to zero-shot generate executable rewards.

---
**Environment: Tetris**

The Tetris environment from Gymnasium's Atari Learning Environment presents the agent with a grid-based game where tetrominoes fall from the top of the screen. The agent's objective is to manipulate and place these tetrominoes to form complete horizontal lines, which are then cleared from the grid. The environment provides visual observations of the game state and discrete actions corresponding to tetromino movements and rotations. The primary task is to encourage

---

the agent to maximize the number of lines cleared over an episode. The current reward signal is sparse, giving positive feedback only when lines are cleared.

**Environment: Snake**

The Snake environment from the Gym-Snake repository provides a grid-based game where the agent controls a snake that moves around the screen to consume randomly placed food items. Each time the snake eats food, it grows in length, increasing the complexity of navigation. The agent receives visual observations representing the current game grid, including the snake's position and the location of the food. The action space is discrete, allowing the agent to choose directional movements (up, down, left, right). The objective is to maximize the length of the snake while avoiding collisions with the walls or the snake's own body. The reward structure is sparse, giving positive reward when the snake consumes food, and a negative reward is given when a collision occurs (episode ends).

**Environment: Flappy Bird**

Flappy Bird is a simple but challenging side-scrolling arcade game in which the player controls a bird that moves continuously to the right. In the actual implementation, the player's x position is fixed while the environment (pipes and background) continuously moves to the left.The only control is to "flap" (making the bird ascend briefly) or do nothing, allowing gravity to pull it down. Vertical movement is automatic when no action is taken. The objective is to navigate the bird through gaps between vertically-aligned pipes without colliding with them or the boundaries of the screen. Each successful pass through a pair of pipes increments the score by one. Colliding with a pipe, the ground, or ceiling ends the game.

**Environment: Humanoid**

The Humanoid environment from Gymnasium's MuJoCo environments provides a 3D bipedal robot simulation designed to mimic human locomotion. The agent controls a humanoid robot with a torso (abdomen), a pair of legs and arms, and tendons connecting the hips to the knees. Each leg consists of three body parts (thigh, shin, foot), and each arm consists of two body parts (upper arm, forearm). The agent receives continuous observations representing joint positions, velocities, center of mass information, inertial data, and external forces. The action space is continuous, allowing the agent to apply torques at 17 different hinge joints. The primary objective is to prevent the humanoid from falling while moving forward as much as possible.

B.3    FUNCTION INTERFACE

FORGE isolates the reward planning from reward generation, prompting LLM with a function interface to generate executable code after a rewards specification is proposed. The function interface is defined for each environment and is enforced at the source code level. This design promotes automatic error-detection and excludes the need for extensive adaptation of the environment source code.

**Interface: Tetris**

```
Args:
    - `action` (int):
        The action taken by the agent.
        For each column on the board, the agent can rotate the
            tetromino counter-clockwise for 0, 1, 2, or 3 times. This
            results in a total of board_width*4 possible actions.
        Therefore, the action space is a Discrete space with
            board_width*4 possible actions. The value is interpreted as
             column index + number of rotations.
        So the actions [0, 1, 2, 3] correspond to the first column and
             the tetromino rotated 0, 1, 2, 3 times respectively.
```

```
            The actions [4, 5, 6, 7] correspond to the second column and
                the tetromino rotated 0, 1, 2, 3 times respectively, and so
                on.
            Action not within the action space is invalid and will result
                in a reward of -1.
    - `curr_board` (2D numpy array): A binary array representation of
        the game board after the `action` is taken, where `1` indicates
         a filled cell and `0` indicates an empty cell. This
        representation also includes the newly apperaed tetromino to be
         placed.
    - `curr_active_tetromino` (2D numpy array): A binary array of the
         same shape as `curr_board`, containing ONLY the tetromino to be
          placed for the NEXT step. Therefore, current board without
        active tetromino is curr_board - curr_active_tetromino.
    - `prev_board` (2D numpy array): A binary array of the same shape
        as `curr_board`, representing the game board of previous step.
    - `prev_active_tetromino` (2D numpy array): A binary array of the
        same shape as `prev_board`. Previous board without active
        tetromino is prev_board - prev_active_tetromino.
    - `lines_cleared` (int): The number of lines cleared resulted from
         the `action` taken in the current step.

Example:
    Consider a 7x5 board at a given point during gameplay, where the
        following inputs are given:
    - action: 9 (column index 2, rotates counter-clockwise for 1 time)
    - curr_board:
        [
            [0, 0, 1, 1, 0],
            [0, 0, 1, 1, 0],
            [0, 0, 0, 0, 0],
            [0, 0, 0, 0, 0],
            [0, 0, 0, 0, 0],
            [0, 0, 1, 1, 0],
            [0, 0, 0, 1, 1]
        ]
    - curr_active_tetromino:
        [
            [0, 0, 1, 1, 0],
            [0, 0, 1, 1, 0],
            [0, 0, 0, 0, 0],
            [0, 0, 0, 0, 0],
            [0, 0, 0, 0, 0],
            [0, 0, 0, 0, 0],
            [0, 0, 0, 0, 0]
        ]
    - prev_board:
        [
            [0, 0, 0, 1, 0],
            [0, 0, 1, 1, 0],
            [0, 0, 1, 0, 0],
            [0, 0, 0, 0, 0],
            [0, 0, 0, 0, 0],
            [0, 0, 0, 0, 0],
            [0, 0, 0, 0, 0]
        ]
    - prev_active_tetromino:
        [
            [0, 0, 0, 1, 0],
            [0, 0, 1, 1, 0],
            [0, 0, 1, 0, 0],
            [0, 0, 0, 0, 0],
            [0, 0, 0, 0, 0],
```

```
                    [0, 0, 0, 0, 0],
                    [0, 0, 0, 0, 0]
                ]
        - lines_cleared: 0

    This example shows that by taking action=9, the game board
        transitions from prev_board to curr_board by placing a z-shape
        tetromino to the lower right corner of the board.
    Additionally, a new tetromino appears at the top of the curr_board
        .
    Since no lines are cleared, the number of lines cleared is 0.
    The example is simplified for clarity. The actual dimension of the
        board is 20x10.

Returns:
    You need to return the reward signal (float) based on the given
        inputs.
```

**Interface: Snake**

```
Args:
    - `game_grid` (2D numpy array): An array representation of the
        current game grid, where `0` indicates an empty cell, `1`
        indicates a food item, `2` indicates a snake body, and `3`
        indicates a snake head.
      The grid follows typical numpy array indexing, i.e. [0,0] is
            located at the upper left most pixel, [0, 1] is the pixel to
            the right of [0,0], [1, 0] is the pixel below [0,0].
    - `prev_game_grid` (2D numpy array): An array representation of
        the game grid on the previous step. prev_game_grid has the same
         shape as `game_grid`.
    - `action` (int): The action taken by the snake that led to the
        current game grid. The action space is discrete, with the
        following possible values: 0-Move up, 1-Move right, 2-Move down
        , 3-Move left.
    - `food_eaten` (bool): Whether the snake has eaten food in the
        current step.
    - `snake_death` (bool): Whether the snake has died in the current
        step.
    - `snake_steps` (int): The number of steps the snake has taken
        since the start of the episode.

Example:
    Consider a 5x5 game grid where the snake has moved only once since
        the start of the episode, the current arguments are:
    - game_grid:
        [
            [0, 0, 0, 0, 0],
            [0, 2, 0, 0, 0],
            [0, 2, 3, 0, 0],
            [0, 0, 0, 0, 0],
            [0, 0, 0, 1, 0]
        ]
    - prev_game_grid:
        [
            [0, 2, 0, 0, 0],
            [0, 2, 0, 0, 0],
            [0, 3, 0, 0, 0],
            [0, 0, 0, 0, 0],
            [0, 0, 0, 1, 0]
        ]
    - action: 1
```

```
    - food_eaten: False
    - snake_death: False
    - snake_steps: 1

    This example shows a snake of length 3, whose body is currently at
        position (1, 1), (2, 1), and the head is at position (2, 2).
        The food is located at position (4, 3).
    The snake took action 1 (move right) from the previous step where
        the snake was at position (0, 1), (1, 1), and the head was at
        position (2, 1).
    Since the snake has not eaten food in the current step, `
        food_eaten` is False.
    Since the snake has not died in the current step, `snake_death` is
         False.
    Since the snake has moved only once since the start of the episode
        , `snake_steps` is 1.

Returns:
    You need to return the reward signal for the current step.
```

**Interface: Flappy Bird**

```
Args:
    - `last_pipe_x` (float): The horizontal position of the last pipe.
    - `last_top_pipe_y` (float): The vertical position of the last top
         pipe.
    - `last_bottom_pipe_y` (float): The vertical position of the last
        bottom pipe.
    - `next_pipe_x` (float): The horizontal position of the next pipe.
    - `next_top_pipe_y` (float): The vertical position of the next top
         pipe.
    - `next_bottom_pipe_y` (float): The vertical position of the next
        bottom pipe.
    - `next_next_pipe_x` (float): The horizontal position of the next
        next pipe.
    - `next_next_top_pipe_y` (float): The vertical position of the
        next next top pipe.
    - `next_next_bottom_pipe_y` (float): The vertical position of the
        next next bottom pipe.
    - `player_y` (float): The vertical position of the player.
    - `player_y_velocity` (float): The vertical velocity of the player
        .
    - `player_rotation` (float): The rotation of the player.
    - `player_x` (float): The horizontal position of the player.
    - `player_width` (float): The width of the player.
    - `player_height` (float): The height of the player.
    - `screen_width` (float): The width of the screen.
    - `screen_height` (float): The height of the screen.

NOTE: All the above arguments are un-normalized. The observation that
    the policy network will receive is normalized.
        The normalization is done as follows:
          - for all x values, normalized_x = x / screen_width
          - for all y values, normalized_y = y / screen_height
          - `player_y_velocity` is normalized as `player_y_velocity` /=
              PLAYER_MAX_VEL_Y, where PLAYER_MAX_VEL_Y=10.
          - `player_rotation` is normalized as `player_rotation` /= 90.
        This note is only for your information. You do not need to
            compute normalized values as it will be done automatically.

Returns:
```

```
You need to return the reward signal for the current step.
```

**Interface: Humanoid**

```
Objective: Obtain the highest score possible, where the exact score
    composition is defined in the `score_info` dictionary.

Args:
  - `score_info` (dict): Dictionary containing the score components:
    - `healthy_reward` (float): A reward is given if the Humanoid is
        alive (Humanoid is alive if the z-coordinate of the torso (the
        height) is in the closed interval given by the healthy_z_range)
        .
    - `forward_reward` (float): A reward for moving forward, this
        reward would be positive if the Humanoid moves forward (in the
        positive x direction / in the right direction).
    - `ctrl_cost` (float): A negative reward to penalize the Humanoid
        for taking actions that are too large.
    - `contact_cost` (float): A negative reward to penalize the
        Humanoid if the external contact forces are too large.
  - `action` (np.ndarray): Action vector of shape (17,) containing
      torques applied to each joint with values constrained to [-0.4,
      0.4]. Actions correspond to:
    0: Torque applied on the hinge in the y-coordinate of the abdomen
        (N m)
    1: Torque applied on the hinge in the z-coordinate of the abdomen
        (N m)
    2: Torque applied on the hinge in the x-coordinate of the abdomen
        (N m)
    3: Torque applied on the rotor between torso/abdomen and the right
        hip (x-coordinate) (N m)
    4: Torque applied on the rotor between torso/abdomen and the right
        hip (z-coordinate) (N m)
    5: Torque applied on the rotor between torso/abdomen and the right
        hip (y-coordinate) (N m)
    6: Torque applied on the rotor between the right hip/thigh and the
        right shin (N m)
    7: Torque applied on the rotor between torso/abdomen and the left
        hip (x-coordinate) (N m)
    8: Torque applied on the rotor between torso/abdomen and the left
        hip (z-coordinate) (N m)
    9: Torque applied on the rotor between torso/abdomen and the left
        hip (y-coordinate) (N m)
    10: Torque applied on the rotor between the left hip/thigh and the
        left shin (N m)
    11: Torque applied on the rotor between the torso and right upper
        arm (coordinate -1) (N m)
    12: Torque applied on the rotor between the torso and right upper
        arm (coordinate -2) (N m)
    13: Torque applied on the rotor between the right upper arm and
        right lower arm (N m)
    14: Torque applied on the rotor between the torso and left upper
        arm (coordinate -1) (N m)
    15: Torque applied on the rotor between the torso and left upper
        arm (coordinate -2) (N m)
    16: Torque applied on the rotor between the left upper arm and
        left lower arm (N m)
  - `prev_action` (np.ndarray): Action vector of shape (17,)
      containing actions on the previous step.
  - `x_coord` (float): The x-coordinate of the torso.
  - `y_coord` (float): The y-coordinate of the torso.
```

```
- `prev_x_coord` (float): The x-coordinate of the torso on the
  previous step.
- `prev_y_coord` (float): The y-coordinate of the torso on the
  previous step.
- `distance_from_origin` (float): The distance from the origin
- `prev_distance_from_origin` (float): The distance from the origin
  on the previous step.
- `healthy_z_range` (tuple of 2 floats): The closed interval of the
  height that the Humanoid is considered alive.
- `obs` (np.ndarray): Observation vector of shape (45, ), containing
    position and velocity information:
  0: z-coordinate of the torso (center) (m)
  1: w-orientation of the torso (center) (rad)
  2: x-orientation of the torso (center) (rad)
  3: y-orientation of the torso (center) (rad)
  4: z-orientation of the torso (center) (rad)
  5: z-angle of the abdomen (in lower_waist) (rad)
  6: y-angle of the abdomen (in lower_waist) (rad)
  7: x-angle of the abdomen (in pelvis) (rad)
  8: x-coordinate of angle between pelvis and right hip (in
     right_thigh) (rad)
  9: z-coordinate of angle between pelvis and right hip (in
     right_thigh) (rad)
  10: y-coordinate of angle between pelvis and right hip (in
      right_thigh) (rad)
  11: angle between right hip and the right shin (in right_knee) (
      rad)
  12: x-coordinate of angle between pelvis and left hip (in
      left_thigh) (rad)
  13: z-coordinate of angle between pelvis and left hip (in
      left_thigh) (rad)
  14: y-coordinate of angle between pelvis and left hip (in
      left_thigh) (rad)
  15: angle between left hip and the left shin (in left_knee) (rad)
  16: coordinate-1 (multi-axis) angle between torso and right arm (
      in right_upper_arm) (rad)
  17: coordinate-2 (multi-axis) angle between torso and right arm (
      in right_upper_arm) (rad)
  18: angle between right upper arm and right_lower_arm (rad)
  19: coordinate-1 (multi-axis) angle between torso and left arm (in
       left_upper_arm) (rad)
  20: coordinate-2 (multi-axis) angle between torso and left arm (in
       left_upper_arm) (rad)
  21: angle between left upper arm and left_lower_arm (rad)
  22: x-coordinate velocity of the torso (centre) (m/s)
  23: y-coordinate velocity of the torso (centre) (m/s)
  24: z-coordinate velocity of the torso (centre) (m/s)
  25: x-coordinate angular velocity of the torso (centre) (rad/s)
  26: y-coordinate angular velocity of the torso (centre) (rad/s)
  27: z-coordinate angular velocity of the torso (centre) (rad/s)
  28: z-coordinate of angular velocity of the abdomen (in
      lower_waist) (rad/s)
  29: y-coordinate of angular velocity of the abdomen (in
      lower_waist) (rad/s)
  30: x-coordinate of angular velocity of the abdomen (in pelvis) (
      rad/s)
  31: x-coordinate of the angular velocity of the angle between
      pelvis and right hip (in right_thigh) (rad/s)
  32: z-coordinate of the angular velocity of the angle between
      pelvis and right hip (in right_thigh) (rad/s)
  33: y-coordinate of the angular velocity of the angle between
      pelvis and right hip (in right_thigh) (rad/s)
```

```
        34: angular velocity of the angle between right hip and the right
            shin (in right_knee) (rad/s)
        35: x-coordinate of the angular velocity of the angle between
            pelvis and left hip (in left_thigh) (rad/s)
        36: z-coordinate of the angular velocity of the angle between
            pelvis and left hip (in left_thigh) (rad/s)
        37: y-coordinate of the angular velocity of the angle between
            pelvis and left hip (in left_thigh) (rad/s)
        38: angular velocity of the angle between left hip and the left
            shin (in left_knee) (rad/s)
        39: coordinate-1 (multi-axis) of the angular velocity of the angle
             between torso and right arm (in right_upper_arm) (rad/s)
        40: coordinate-2 (multi-axis) of the angular velocity of the angle
             between torso and right arm (in right_upper_arm) (rad/s)
        41: angular velocity of the angle between right upper arm and
            right_lower_arm (rad/s)
        42: coordinate-1 (multi-axis) of the angular velocity of the angle
             between torso and left arm (in left_upper_arm) (rad/s)
        43: coordinate-2 (multi-axis) of the angular velocity of the angle
             between torso and left arm (in left_upper_arm) (rad/s)
        44: angular velocity of the angle between left upper arm and
            left_lower_arm (rad/s)
    - `prev_obs` (np.ndarray): Observation vector of shape (45, ),
        containing position and velocity information on the previous step
        .
    - `cfrc_ext` (dict): Dictionary containing the external contact
        forces on the body parts. Each body part is a vector that
        specifies force x,y,z and torque x,y,z.
        - `torso` (np.ndarray): External contact forces on the torso.
        - `lwaist` (np.ndarray): External contact forces on the lwaist.
        - `pelvis` (np.ndarray): External contact forces on the pelvis.
        - `right_thigh` (np.ndarray): External contact forces on the right
            thigh.
        - `right_shin` (np.ndarray): External contact forces on the right
            shin.
        - `right_foot` (np.ndarray): External contact forces on the right
            foot.
        - `left_thigh` (np.ndarray): External contact forces on the left
            thigh.
        - `left_shin` (np.ndarray): External contact forces on the left
            shin.
        - `left_foot` (np.ndarray): External contact forces on the left
            foot.
        - `right_upper_arm` (np.ndarray): External contact forces on the
            right upper arm.
        - `right_lower_arm` (np.ndarray): External contact forces on the
            right lower arm.
        - `left_upper_arm` (np.ndarray): External contact forces on the
            left upper arm.
        - `left_lower_arm` (np.ndarray): External contact forces on the
            left lower arm.
    - `prev_cfrc_ext` (dict): Dictionary containing the external contact
        forces on the body parts on the previous step.
    - `terminated` (bool): Whether the episode has terminated due to the
        humanoid falling.
    - `truncated` (bool): Whether the episode was truncated due to
        reaching the maximum timestep limit.

Returns:
    A float representing the custom reward signal for the current step.
```

# C    PROMPTS

This section presents the prompts used in FORGE, in the following order:

1. **Reward Planner System Prompt**
2. **Reward Engineer System Prompt**
3. **Reward Planner Reward Initialization Planning Prompt**
4. **Reward Engineer Reward Crossover Prompt**

---

**Planner System Prompt**

```
You are an experienced AI researcher. You need to design the reward
structure for training reinforcement learning agents.

Return a list of individual reward components that could be included
in a dense reward structure. Consider both diversity and clarity in
your design. For each component, you should:
    - Be specific and detailed, describing exactly what behavior it
    encourages.
    - Be suitable for generating a real-time reward signal at every
    time step.
    - Do not assume the actual setup of the environment. This will be
    provided when implementing the reward.

When implementing the reward function, you should:
    - Adhere exactly to the given function header without changing its
     name, parameters, or structure.
    - Only implement the functionality specified in the docstring. Do
    not modify the docstring.
    - If the function header lacks necessary information to implement
    the reward function (e.g., missing state details, actions, or
    environmental context), clearly state why the implementation
    cannot proceed and abort the task.
    - If implementation is possible, return the complete code wrapped
    in markdown-style Python code block ('''python and ''').
    - You can define helper functions inside the main function, but
    your final code should include only one function overall.
```

---

**Engineer System Prompt**

```
You are a code generation assistant specialized in reinforcement
learning (RL). Your task is to generate a Python function that
implements reward function for training an RL agent.

You must follow the rules below:
    - Return the complete code wrapped in markdown-style Python code
    block ('''python and ''').
    - You can define helper functions inside the main function, but
    your final code should include only one function overall.
    - Write out the code explicitly if using given functions.
```

---

**Reward Initialization Planning Prompt**

```
Design the reward structure for training an reinforcement learning
agent based on the environment specification:

<Environment>{env_description}</Environment>
```

```
Format your response as a list and return as many reward components as
 possible to facilitate agent exploration.
```

**Reward Crossover Prompt**

```
Consider the reinforcement learning environment:

<Environment>{env_description}</Environment>

Carefully examine the following reward components ({reward1} and {
reward2}) and their respective training results:

<Result1>{result1}</Result1>

<Result2>{result2}</Result2>

Please design a new reward component that is a combination of the two
given components, taking into account their respective training
results. The new function should have a clear and concise docstring
that explains what the new reward component will do. Additionally,
make sure to keep the implementation self-contained without adding any
 extra or redundant functionality. Do not change the function
arguments. If the implementation is not possible, explain the reason.
Return the complete code wrapped in markdown-style Python code block
(```python and ```).
```

