# OpenReview forum: "Forging Better Rewards: A Multi-Agent LLM Framework for Automated Reward Evolution"
_ICLR.cc/2026/Conference — ICLR 2026 Conference Withdrawn Submission_

### Official Review · Reviewer_dwcw · 2025-10-30

**Soundness:** 2
**Presentation:** 2
**Contribution:** 1
**Rating:** 2
**Confidence:** 5

**Summary:**

This paper introduces FORGE (Feedback-Optimized Reward Generation and Evolution), a multi-agent framework that automates reward synthesis for reinforcement learning using large language models (LLMs). FORGE replaces traditional genetic algorithm encodings with LLM-guided crossover, uses a planner-based zero-shot initialization to generate structured reward functions, and introduces a depth metric to quantify reward complexity over evolutionary iterations. A reward pool is maintained as memory to manage and refine candidates. Experiments are conducted across four environments; three games (Tetris, Snake, Flappy Bird) and a continuous-control task (MuJoCo Humanoid), showing performance improvements over prior methods such as Eureka and REvolve, while claiming enhanced stability, interpretability, and token efficiency.

**Strengths:**

1.1 The paper compares different LLMs and evaluates on multiple environments.

1.2 The experiments are broad and include several baselines.

1.3 The depth measurement is a welcome addition, giving a way to track reward composition complexity.

**Weaknesses:**

2.1 The comparison with Eureka is unfair. Eureka is greedy (keeps only the best reward per generation), not population-based. Plotting average population scores for FORGE and REvolve, but not the best-per-generation for Eureka, makes the figure incomparable. A fair comparison would show the maximum score for each generation across all methods.

2.2 The REvolve baseline results are incorrect. The authors state that they use the environment's extrinsic rewards as the fitness score. However, in Fig. 2, the population average for REvolve decreases over generations, contradicting REvolve’s framework, which adds individuals only if their fitness score exceeds the current population average. This guarantees that the average will never decrease (section 3.4 of the REvolve Paper).

2.3 The table results for the Humanoid task do not match the video output. The task is to move as fast as possible on the x-axis without falling. REvolve’s humanoid performs this correctly and visibly better, while FORGE’s agent moves much worse, although achieving high reward results. The results and videos are thereby inconsistent.

2.4 The claimed “stability”, “interpretability”, and “token efficiency” are not supported. Figure 2 shows dips even for FORGE, so it is not stable. “Interpretability” is only a depth-vs-score correlation and does not explain why rewards work. “Token efficiency” is argued but not tested.

2.5 The authors claim that passing only the reward function and best score is sufficient and that raw metrics are not beneficial. Metrics can indicate which reward components failed and guide improvement in the next generation. REvolve showed that better quality feedback leads to better performance. No results are showed to support that claim.

2.6 Line 217-219: the authors say “to address these limitations we generalize the evolutionary process by incorporating LLM inference…”. This is presented as if it is new, which is not (T2R/Eureka were first).

2.7 It is unclear to me what exact rule is used to retain or discard individuals and when/how the pool is pruned.

2.8 The Planner can be viewed as structured prompt design rather than actual planning. It is only used for zero-shot initialization, not iterative optimization.

**Questions:**

3.1 Can the authors explain why the average population scores for REvolve decrease over generations, even though REvolve’s framework guarantees a non-decreasing average as described in the original paper?

3.2 Can the authors explain why they plot average population scores instead of the best-per-generation values, and provide the additional results showing the best scores per generation for all methods to make the comparison fair?

3.3 Can the authors elaborate on the points raised in 2.6 and 2.7?

---

> ### Author Response · Authors · 2025-11-20
>
> Thank you. We give point-to-point replies to your questions:
>
> ---
>
> **Q1:** The comparison with Eureka is unfair. Plotting average population scores for FORGE and REvolve makes the figure incomparable. A fair comparison would show the maximum score for each generation across all methods.
>
> **A1:**  After plotting the maximum score for each generation across all methods, we found that Eureka underperforms even more evidently across 3 out of the 4 environments (except for Snake, which appears tied). Specifically, in three environments, Eureka best-per-generation scores are below REvolve/Forge on almost all generations.
>
> It is true that Eureka is based on greedy strategy, but greedy solution generally is not optimal. We have specified in Section 4.2 that for Eureka we increase the number of functions to generate to 16, giving Eureka more chances to sample **additional** functions. So the reason we adopt average population score is exactly to make the comparison fair for REvolve/Forge since they adopt more conservative strategy that involves combining previous rewards.
>
> ---
>
> **Q2:** The REvolve baseline results are incorrect. In Fig. 2, the population average for REvolve decreases over generations, contradicting REvolve’s framework, which adds individuals only if their fitness score exceeds the current population average. This guarantees that the average will never decrease (section 3.4 of the REvolve Paper).
>
> **A2:**  We quote from REvolve paper: "Individuals are retained if they contribute to an increase in the average fitness of their respective **sub-populations**". The underperforming individuals are not added to this **sub-populations** ("islands" in terms of evolutionary algorithm), which differ from the notion of per-generation-population that take into account all generated reward function at each generation round. The evolutionary algorithm adopted by REvolve ensures average "islands" population scores always increase, but does not prevent the LLM from generating bad functions.
>
> We want to emphasize that Table 2 reports the best scores of **ALL** reward functions generated for the corresponding method, and the average scores plotted in Figure 2 are scores per-generation-population.
>
> ---
>
> **Q3:** The table results for the Humanoid task do not match the video output. The task is to move as fast as possible on the x-axis without falling. REvolve’s humanoid performs this correctly and visibly better, while FORGE’s agent moves much worse, although achieving high reward results. The results and videos are thereby inconsistent.
>
> **A3:**  The difference in trained RL agent performance is due to use of different RL algorithm. REvolve uses SAC for humanoid that is much slower, as reported in their paper, it takes 24 hours on 10 A100 GPU to complete a single generation. We uses PPO which is much faster (~1.8hrs to complete a single generation), with a cost of obtaining less gain per training step. The choice of algorithms has limited effect on LLM generation since this information is not exposed in prompt, and REvolve actually uses different algorithms (DDQN on autonomous driving and SAC on MuJoCo).
>
> We believe a better scenario to question the video demonstration will be on Eureka, which demonstrated, for the first time, that LLM can surpass human-engineered rewards in complex low-level manipulation tasks. However, in REvolve/Forge, focusing on demonstration undermines the purpose of comparing LLM generated reward. Additionally, using SAC, we obtain a visibly much better humanoid in just under 1M training steps (7k scores compared to 1k scores using PPO with 5M steps, and the video already demonstrates humanoid running without falling).
>
> ---
>
> **Q4:** The claimed “stability”, “interpretability”, and “token efficiency” are not supported. Figure 2 shows dips even for FORGE, so it is not stable. “Interpretability” is only a depth-vs-score correlation and does not explain why rewards work. “Token efficiency” is argued but not tested.
>
> **A4:** Since we average scores across per-generation-population, we expect to have dip because the quality of LLM generated code is not guaranteed. Depth itself is an interpretability metric because it is not obtainable from other methods. We already show comparison of token consumption in the paper. Please elaborate on the meaning of "tested".

---

> > ### Author Response · Authors · 2025-11-20
> >
> > **Q5:** The authors claim that passing only the reward function and best score is sufficient and that raw metrics are not beneficial. Metrics can indicate which reward components failed and guide improvement in the next generation. REvolve showed that better quality feedback leads to better performance. No results are showed to support that claim.
> >
> > **A5:**  We want to respectfully ask the reviewer to indicate where in the paper have we specified that raw metrics are not beneficial. The only claim we have made is that our method shows better performance on the environments we have tested. When producing results using REvolve, we adopt the exact same strategy in REvolve to generate score metrics for each reward component and the experimental result is shown in Table 2. When saying we claimed that passing best score is sufficient, please elaborate on what aspects do you find us claim this sufficiency.
> >
> > ---
> >
> > **Q6:** Line 217-219: the authors say “to address these limitations we generalize the evolutionary process by incorporating LLM inference…”. This is presented as if it is new, which is not (T2R/Eureka were first).
> >
> > **A6:**  We have acknowledged in Section 1 and Section 4.2 the existing works that leverage LLMs to generate executable code as reward functions. If our wording has given the notion of making false claims, we will definitely make correction. On the other hand, we have to respectfully ask that does listing this single phrase contribute to any constructive advice?
> >
> > ---
> >
> > **Q7:**  It is unclear to me what exact rule is used to retain or discard individuals and when/how the pool is pruned.
> >
> > **A7:**  The reward pool has a fixed size which is a hyperparameter (32 in our experiments). The pruning process happens at the end of each generation round: We check whether the size limit is exceeded after adding each individual. If it is true, we remove the reward with the lowest score.
> >
> > ---
> >
> > **Q8:**  The Planner can be viewed as structured prompt design rather than actual planning. It is only used for zero-shot initialization, not iterative optimization.
> >
> > **A8:**  The Planner is designed to explore the reward space and propose as diverse reward structures as possible. Compared to Eureka which provide environment code, Forge only provide textual description of the environment and task in the planning phase to facilitate more explorative planning. The actual implementation is done by Engineer. In our view, the "structured prompt design" achieves the work of planning through LLM reasoning. Please elaborate on what is lacking from doing an "actual planning". Indeed, a more sophisticated and better design is to also involve LLM in dynamic planning and reasoning in the evolutionary process, and we believe that is a promising future direction.

---

> > ### Comment · Reviewer_dwcw · 2025-11-20
> >
> > Thank you for the clarification:
> >
> > A1: The statement that “after plotting the maximum score per generation Eureka underperforms even more” is unclear. It is mathematically impossible for the maximum score of a given set to be worse than its own average. Could you clarify what you mean by this? In addition, plotting the average score for Eureka is difficult to interpret, since Eureka is a greedy algorithm that retains only the single best reward function from the previous round and does not maintain or evolve a population. Averaging over raw candidates that Eureka immediately discards does not correspond to the behavior of the algorithm and therefore does not seem to be a meaningful metric for comparison.
> >
> >
> > A2: If I understand your response correctly, REvolve’s selection rule is implemented properly in your code (i.e., individuals below the current population mean are not added to the evolving sub-population). However, Figure 2 does not plot the fitness of this retained population. Instead, the figure reports the average fitness of all reward functions generated in that iteration, including those that are immediately rejected and never become part of REvolve’s population.
> > If REvolve never keeps the bad individuals, why are you comparing using averages that include those bad individuals? What purpose does that serve? Could you clarify the motivation for using this metric given that it does not represent the behavior of REvolve’s evolving population?
> >
> >
> > A3,A4: The explanation based on PPO vs. SAC is difficult to interpret. PPO is indeed cheaper per update, but it is also substantially less sample-efficient than SAC. It typically requires more training steps to reach comparable performance, not fewer. Since the paper does not report the training-step budget or runtime for PPO, the statement that a generation “converges” in 1.8 hours is difficult to evaluate.
> > In addition, Table 2 reports PPO results for all methods, and the video demonstration also corresponds to PPO. The SAC results you cite in the rebuttal (e.g., “7k vs 1k”) do not appear anywhere in the paper, nor is any SAC-trained humanoid shown. Because my original concern is about the inconsistency between the PPO reward reported in Table 2 and the PPO behavior shown in the video, references to SAC performance do not resolve the issue.
> > If SAC is used to support the claim that FORGE achieves substantially better humanoid behavior, could you provide the corresponding SAC video or quantitative results?
> > I do not see how “depth” provides interpretability. Likewise, dips in Figure 2 are the opposite of stability.
> >
> >
> >
> > A5: I understand that the paper does not explicitly claim that raw metrics are “not beneficial.” My point is about the design of FORGE: in Sections 3.1–3.2, the only feedback passed to the LLM during evolution is the reward function itself and a single scalar extrinsic score  J*. This means the LLM receives no information about which reward terms underperformed, or became redundant, etc. So, as far  as I understand the LLM evolves "blindly".
> >
> >
> > A6: I "flagged" this sentence because the wording “we generalize the evolutionary process by incorporating LLM inference” reads as if the use of LLMs for reward evolution is introduced here for the first time. Prior work such as Eureka and T2R already incorporated LLM inference into reward design, so this phrasing overstates novelty. My comment was only to suggest clarifying the wording to avoid this implication.

---

> > > ### Author Response · Authors · 2025-11-20
> > >
> > > Please see the following for further clarification:
> > >
> > > ---
> > >
> > > **AA1:**  The statement “after plotting the maximum score per generation Eureka underperforms even more” means Eureka underperform REvolve/Forge when plotting the maximum score for each generation: max(Eureka) < max(Revolve/Forge), in 3 out the 4 environments except snake, which apperes tied. The score differences are even larger than the score differences shown in Figure 2. The following table shows the maxium scores (normalized across environments):
> > >
> > > Snake/Humanoid
> > > | |round 0|round 2|round 4| round 6| round 8| round 10|
> > > | ----------- | :-----------: |:-----------:| :-----------:| :-----------:|:-----------:|:-----------:|
> > > |Forge  | 0.79/0.77| 0.74/0.75| 0.84/0.78| 1.0/0.84 | 0.71/0.81| 0.70/1.0|
> > > |REvolve| 0.71/0.64| 0.74/0.63| 0.97/0.67| 0.71/0.83| 0.68/0.70| 0.71/0.64|
> > > |Eureka | 0.52/0.05| 0.83/0.71| 0.75/0.56| 0.76/0.62| 0.80/0.63| 0.80/0.61|
> > >
> > > For clarity, odd-numbered rounds are excluded and only snake and humanoid are shown.
> > >
> > > When plotting the average scores for Eureka, we also plotted the average scores for REvolve/Forge, which also include raw candidates that do not perform well and can be immediately discarded. The point is to make the comparison **fair**, as you questioned in your first response. If you still think this comparison is not meaningful, the above table leads to the same conclusion, where we plotted maxium scores of each round just like you required.
> > >
> > > ---
> > >
> > > **AA2:** Sure, I think an equivalent statement of your point is Figure 3 in the REvolve paper, which show the fitness score never decreases. Please correct me if my understanding is wrong: notice how the fitness scores plateaued in generation round > 5, showing that the best reward function has already appeared. We can present our result the same way, plot the score of the best-performing reward function in the population, and only add to the population if a better reward function appears. This will make the plot looks similar to Figure 3 in REvolve paper. Actually, Figure 2 in our paper has already revealed this purpose, if you just plot points of the same height after where the maxium scores occurred.
> > >
> > > We believe Figure 2 contain more information that it shows the **stability** of the algorithm. The intuition is that the algorithm can discover a surprisingly good reward function by chance, which will appear as a point high up on the plot that it looks like an outlier. This is very possible due to the inconsistency of LLM response. However, if we also include the scores of bad individuals when showing the result, it will tell a rough sense of how **stable** the algorithm is.
> > >
> > > Therefore, you may argue that our algorithm is unstable in the Snake environment shown in Figure 2, because the score fluctuation is large and the result may not appear as good if we re-run the experiment.
> > >
> > > ---
> > >
> > > **AA3,AA4:** The SAC results we reported in the rebuttal is a small experiment to show that switching RL algorithm can boost RL policy performance, and it is not in the paper. The original experiments using PPO is indeed not enough to say the training converged, and, like you said, the training steps (5M) might be too few. Our original motivation is to optimize training time since the performance of RL policy is not the main concern. We are running humanoid experiments using SAC, but each generation round take more than 24hrs. We will post the result here and in the revised paper once the experiments complete.
> > >
> > > Please see AA2 for the explanation of plotting, and it agrees with your claim that the algorithm is not stable, so are the baselines. As long as LLM generation is inconsistent, this problem will persist. We can present the result as explained in AA2, but it does not solve the actual problem of instability.
> > >
> > > As explained with other reviewer: Depth measure provides an intuitive interpretation of LLM generated reward structure and a clear roadmap for the discovering of optimal rewards (Figure 4 and Figure 5). It can be seen from Figure 3 (a) that, for game environments, the maxium score has already occurred at a depth of around 3-4 even though more reward offsprings are created to reach even deeper depth. On the other hand, for humanoid environment, the maxium score occurs at a depth of 7. This observation correlates depth measure to the complexity of the environment and the difficulty of finding the optimal reward function, and it has revealed a possibility that we can use much less resources and time to achieve the same result: Instead of running for a fixed number of rounds (in case of FORGE, REvolve and Eureka), a more efficient and dynamic methodology would be to use metrics like depth to guide LLM exploration in the reward space.

---

> ### Author Response · Authors · 2025-11-20
>
> **AA5:** Thank you for the clarification. Our point is to show LLM generation in our framework have shown better experimental result on our selected environments. One can definitely argue its generalizability to more complex, more diverse or even simpler environments. But as far as our experiments show, better result is observed even if "LLM evolves blindly" according to your explanation.
>
> We also want to argue that the evolution process is not a blind process, and it actually incorporate some extent of the raw metrics: The LLM can see the code of reward functions that are required to be combined, and it can reason on whatever metrics are defined in the function itself. Most importantly, as shown in Figure 2 and Table 2, the scenario where LLM truly evolves "blindly" is the baseline of **"context-aware LLMs"**, where we have tested 4 models (Claude 4, Grok 4, o3 and GPT-5) and ask them to simply improve the reward function given all history scores and function code.
>
> ---
>
> **AA6:** Thank you for your clarification. Our intention is definitely to just state the fact that “we generalize the evolutionary process by incorporating LLM inference”. We will make sure to change the wording and check the rest of the paper for possible similar issues.

---

> ### Author Response · Authors · 2025-11-28
>
> Dear Reviewer,
>
> As a supplement to the point **AA3,AA4:**, We have partially completed the experiments on humanoid environment using SAC. Due to time constraint (~24 hours per iteration round and total of 10 rounds and 3 baselines), we set the training steps to be **1M steps**, unlike REvolve, which had run for 5M steps. We report the result in the following table, where the scores are the **MAX** scores of each round:
>
> Humanoid SAC
> | |round 0|round 2|round 4| round 6| round 8| round 10|
> | ----------- | :-----------: |:-----------:| :-----------:| :-----------:|:-----------:|:-----------:|
> |Forge  | 7979| 6288| 6333| **8363**| 7658| 7761|
> |REvolve| 7357| 7954| 7529| **7961**| 7757| 7237|
> |Eureka | 68| 7865| 7678| 7482| 7502| **7999**|
>
> The odd-numbered rounds are excluded for clarity, and the bold-faced score is the maximum score achieved by the corresponding method. Please also see the updated supplementary material for the **video** of the trained RL agent using FORGE.
>
> We acknowledge that we have run fewer training steps than the baseline method, and the result can be inconclusive since the training might not have converged. Full experiments on humanoid environment will be reported once it become available.
>
> Since the deadline is approaching, we want to kindly ask if there is any remaining question? Please also see our response to your concerns about our method, experiments and motivation.

---

### Official Review · Reviewer_P9WS · 2025-10-31

**Soundness:** 3
**Presentation:** 2
**Contribution:** 3
**Rating:** 6
**Confidence:** 3

**Summary:**

This paper proposes the FORGE, a multi-agent LLM framework that combines structured reward initialization, evolutionary refinement, and explicit complexity modeling. The core problem is that manual reward engineering is costly and suboptimal, while existing LLM-based methods (e.g., Eureka, REvolve) often produce unstable or opaque rewards. FORGE employs a Planner agent to generate modular reward specifications from task objectives and environment dynamics, followed by an evolutionary process where rewards are selectively combined and refined using LLM-guided crossover. Key contributions include a reward pool for efficient memory, a depth measure to quantify structural complexity, and token-efficient evolution. Extensive experiments across three games (Tetris, Snake, Flappy Bird) and a robotics task (Humanoid) demonstrate that FORGE achieves significant performance improvements—up to 38.5% over Eureka and 19.0% over REvolve in Humanoid—while maintaining competitive token usage.

**Strengths:**

**Method Design**: FORGE introduces a structured two-stage process (Planner-based initialization and Engineer-driven evolutionary refinement) that moves beyond direct LLM sampling for reward generation. Sec. 3; Fig. 1. This modular approach enhances interpretability and stability compared to prior methods like Eureka and REvolve.

**Comprehensive Experimental Evaluation**: The framework is tested across four distinct environments (three games and one simulated robotics task), covering both discrete and continuous control settings. Sec. 4.1; Table 2. FORGE consistently outperforms baselines, including general agentic frameworks, context-aware LLMs, and native environment rewards.

**Token Efficiency**: Despite performance improvements, FORGE maintains competitive token usage by constraining LLM inference to modifying small subsets of reward functions. Sec. 4.5 This is a critical advantage for scalable deployment compared to other multi-agent LLM frameworks.

**Weaknesses:**

**Limited Generalization**: Evaluation is confined to simulated environments (MuJoCo, games), with no evidence of testing in real-world robotics or complex physical systems (Sec. 5). Tasks lack diversity in observation spaces (e.g., low-dimensional vs. high-dimensional inputs), raising questions about scalability to vision-based or partially observable domains (Table 1). No cross-environment transfer experiments to assess reward function generalization beyond training domains (Sec. 4.1).

**Incomplete Analysis of Evolutionary Components**: The probabilistic selection scheme (Eq. 6) uses unnormalized scores as weights, but no ablation is provided on alternative selection strategies (e.g., rank-based or tournament selection). Crossover operation relies solely on LLM inference without mutation mechanisms, potentially limiting diversity in later generations (Sec. 3.2). Depth measure is defined recursively but lacks theoretical grounding or comparison to other complexity metrics (e.g., code length, entropy) (Eq. 4).

**Questions:**

1.	How does FORGE handle environments with highly sparse or delayed extrinsic rewards, and does the depth measure correlate with performance in such settings? (Sec. 4.1; Fig. 3)
2.	Could the evolutionary process be enhanced by incorporating multi-objective optimization to balance reward complexity and performance, rather than relying solely on extrinsic return? (Eq. 6; Sec. 3.2)
3.	What are the specific failure modes of FORGE in cases where the LLM generates invalid reward functions, and how frequently do these occur across different environments? (Sec. 3.2; Sec. 4.4)

---

> ### Author Response · Authors · 2025-11-20
>
> Thank you for your valuable comments! We answer your questions in the following:
>
> ---
>
> **Q1:**  How does FORGE handle environments with highly sparse or delayed extrinsic rewards, and does the depth measure correlate with performance in such settings?
>
> **A1:**  Thank you for your insightful question, as it points to an overlooked detail in reward shaping with LLM. An observation from Figure 2 is that many baseline methods fail on the zero-shot round for Tetris environment, as the extrinsic reward is highly sparse, only giving reward when a line is cleared in the game. Even with latest LLMs, it is not the case that LLM can propose an optimal reward function in a one-shot manner without a few rounds of testing. This is also true for general agentic frameworks (ChatGPT Agent and MetaGPT) which have access to an abundance of internet resources. FORGE differentiate from other methods that it involves a planning stage, where the LLM is prompted to generate diverse and atomic reward structures. These rewards are then tested individually in the environment to essentially bootstrap the evolutionary process for more efficient exploration later in the evolution rounds. Although, with increased training rounds, most baselines have achieved improved result, it remains to be tested in a more complex and highly sparse environment the effectiveness of this planning strategy. Thank you again for noticing this detail and bringing up the valuable question!
>
> Depth measure provides an intuitive interpretation of LLM generated reward structure and a clear roadmap for the discovering of optimal rewards (Figure 4 and Figure 5). It can be seen from Figure 3 (a) that, for game environments, the maxium score has already occurred at a depth of around 3-4 even though more reward offsprings are created to reach even deeper depth. On the other hand, for humanoid environment, the maxium score occurs at a depth of 7. This observation correlates depth measure to the complexity of the environment and the difficulty of finding the optimal reward function, and it has revealed a possibility that we can use much less resources and time to achieve the same result. Instead of running for a fixed number of rounds (in case of FORGE, REvolve and Eureka), a more efficient and dynamic methodology would be to use metrics like depth to guide LLM exploration in the reward space. To the best of our knowledge, no other works have demonstrated this possibility experimentally in reward shaping with LLM.
>
> ---
>
> **Q2:**  Could the evolutionary process be enhanced by incorporating multi-objective optimization to balance reward complexity and performance, rather than relying solely on extrinsic return?
>
> **A2:**  Thank you so much for your valuable advices. Indeed, our method of relying on LLM inference and a selection mechanism could be non-optimal and may use more resources than necessary, as discussed in Q1. Multi-objective optimization may combine conflicting reward components more effectively and can provide better theoratical support. Like in the REvolve paper, they explicitly require the reward function to output reward metrics in additional to a single value of scores, and feed these metrics as additional information to the LLM. We believe that further optimization can be made with methods that involve multi-objective optimization.
>
> ---
>
> **Q3:** What are the specific failure modes of FORGE in cases where the LLM generates invalid reward functions, and how frequently do these occur across different environments?
>
> **A3:** In cases where LLM generates invalid reward functions, they are simply discarded without asking LLM to regenerate. This design alleviate the cost of token consumption and avoid potential hallucination when asking LLM to fix its errors. It does rely on the LLM's ability to write runnable code in a single query, but the experimental results have proven to be quite successful on Claude-4. Please see the common questions section for a comparison of success rates across two models. The hyperparameters used are exactly the same as specified in Section 4.2.

---

> > ### Author Response · Authors · 2025-11-27
> >
> > Dear Reviewer,
> >
> > Since the deadline is approaching, we are wondering if there are any additional details we can provide to address your concerns. We will gladly answer any further questions if needed.
> >
> > Thank you again for your time and we are looking forward to your response.

---

### Official Review · Reviewer_8Pcf · 2025-11-01

**Soundness:** 2
**Presentation:** 3
**Contribution:** 2
**Rating:** 4
**Confidence:** 3

**Summary:**

FORGE is a multi-agent LLM framework for automated reward evolution, using a Planner agent for structured zero-shot reward initialization, an Engineer agent for iterative selection and crossover refinement, and explicit depth metrics for complexity. Evaluated on Tetris, Snake, Flappy Bird, and Humanoid (MuJoCo), it outperforms baselines like Eureka, REvolve, and context-aware LLMs, achieving up to 38.5% gains over Eureka on Humanoid while maintaining token efficiency.

**Strengths:**

1.Structured initialization via Planner yields strong zero-shot rewards, outperforming direct LLM sampling; evolutionary refinement drives consistent gains across discrete and continuous domains.
2.Experiments show superior performance in both zero-shot and evolved settings, with ablation studies confirming the value of key components like selection and planning.
3.Introduces reward pool and depth metrics for stability, interpretability, and efficiency, enabling analysis of complexity-performance correlations (e.g., optimal depth 3 for games, 7 for Humanoid).

**Weaknesses:**

1.Relies on Claude Sonnet 4 without ablations on other LLMs—results may be model-specific and degrade with open-source alternatives.
2.Claims token efficiency but consumes more in some environments; lacks full compute cost breakdown or scaling analysis for larger tasks.
3.Baselines use the same LLM, but adaptations (e.g., replacing human feedback in REvolve) may not be optimal; multi-agent setup adds overhead without clear justification over simpler methods.
4.Depth analysis is insightful but underexplored—e.g., no explanation for domain-specific optima or robustness to hallucinations in crossover.
5.Limited to simulated environments; no real-world robotics tests, overlooking challenges like sensor noise, delays, and safety that could undermine practicality.

**Questions:**

1.What is the sensitivity to the base LLM? Results with open-source models like Llama or smaller variants?
2.Detailed token/compute costs per iteration/environment? How does efficiency scale to more complex domains?
3.Why does evolution sometimes underperform context-aware LLMs on average? Mechanisms to boost consistency?
4.How can depth metrics guide early stopping? Thresholds or heuristics for optimizing iterations?

---

> ### Author Response · Authors · 2025-11-20
>
> Thank you for your constructive comments! Please see the following responses to your concerns:
>
> ---
>
> **P1:**  1. What is the sensitivity to the base LLM, results with open-source models like Llama or smaller variants?
>
> **A1:**  Please see the common questions section for experimental result on Llama-3.1-70B-Instruct-Turbo. We choose Llama-3.1 for its open-sourced property, release date (July 2024, compared to Claude-4-Sonnet, which was released in May 2025) and weaker benchmarks performance [1]. Based on the experimental results, Forge still discovers better reward functions across all tested environments. We will include this result in a revised version.
>
> ---
>
> **P2:** Detailed token/compute costs per iteration/environment? How does efficiency scale to more complex domains?
>
> **A2:** Our experiments are conducted on 16 Core Intel Emerald Rapids CPUs. If run in parallel, a single round of policy training can be completed within 2 hours. Compared to REvolve, which claim to use 16 NVDIA A100 GPUs and need 24 hours to run a single round, and Eureka, which applies 8 A100 GPUs with GPU-accelerated training, our method require much less computation resources and time.
>
> However, the resources constraint limit us from applying Forge to more complex environments and real-world scenarios. But the generalizability still holds because we have showed that even latest LLM can struggle on relatively simple game environments on the task of reward-shaping, and the performance differs under different frameworks.
>
> Please see the following table for a summary of token consumption per iteration. We report the result on Llama-3.1 for comparison with the token consumption result on Claude-4-Sonnet shown in Figure 7:
>
> | |Humanoid Init/Evo|Tetris Init/Evo|Flappy_bird Init/Evo| Snake Init/Evo|
> | ----------- | :-----------: |:-----------:| :-----------:| :-----------:|
> |Forge  |47903/68872.4| 51846/70503.3|46763/72884.9| 57817/68644.2|
> |REvolve|31495/89519.5| 21488/66141.4|18277/65876.2| 17672/63767.8|
> |Eureka |4142/83247.3 | 2730/71096.1 |2115/57514.3 | 2122/48699.3 |
>
> For clarity, we only report a shortened table here and will include the full result in a revised version. In the above table, **Init** and **Evo** denote the initialization round and all rounds after initialization (evolved), similar to Table 2. The total token consumption for the initialization round is reported as it is, and the total token consumption for evolved rounds is averaged over all later rounds.
>
> As shown in the table, Forge has a limited increase in token consumption in the evolution rounds, with at most ~20K tokens more than Eureka, roughly 0.3 dollar if we are using Claude-4-Sonnet. The large consumption in the initialization round indicates the increasing planning effort for both Forge and REvolve. At ~50K tokens levels the cost is roughly 0.75 dollar.
>
> ---
>
> **P3:**  Why does evolution sometimes underperform context-aware LLMs on average? Mechanisms to boost consistency?
>
> **A3:**  Sorry for causing the confusion. Figure 2 has context-aware LLM and evolution methods on the same plot; the former case reports the score for only a single generated function, but the latter case has scores averaged over the entire generation round. Therefore, the score of a single function generated by context-aware LLM might exceed the average score of a batch of functions generated by evolution methods. Due to the inconsistent nature of LLM generation, there is also chance that LLM can generate a surprisingly good reward function in a zero-shot manner, but our results have shown that evolution methods are almost always better.
>
> Our mechanism to boost consistency is handled by the Engineer agent, which always combine two functions from the function pool instead of asking LLM to mutate existing function or generating new ones. This mechanism reduces chances of generating invalid function but still keep a certain level of exploration.
>
> ---
>
> **P4:**  How can depth metrics guide early stopping? Thresholds or heuristics for optimizing iterations?
>
> **A4:**  Thank you for your great question, as we have already been working on this direction but have not yet obtained results. As can be seen from Figure 3, the game environments have already obtained the highest scores at depth of around 4, indicating that the best reward function can be discovered with as little as only 4 iterations (7 for the more complex humanoid environment). Instead of the selection mechanism with function pool implemented in the paper, a more effective design may involve dynamic LLM planning that guide the exploration.
>
> [1] “Llama 3.1 Instruct 70B: Intelligence, Performance & Price Analysis.” ArtificialAnalysis.ai,
> https://artificialanalysis.ai/models/llama-3-1-instruct-70b.

---

> > ### Author Response · Authors · 2025-11-27
> >
> > Dear Reviewer,
> >
> > Since the deadline is approaching, we are wondering if there are any additional details we can provide to address your concerns. We will gladly answer any further questions if needed.
> >
> > Thank you again for your time and we are looking forward to your response.

---

### Official Review · Reviewer_d2ow · 2025-11-02

**Soundness:** 3
**Presentation:** 3
**Contribution:** 2
**Rating:** 4
**Confidence:** 4

**Summary:**

This paper proposes FORGE, a multi-agent LLM + evolution framework for automated reward function design. FORGE first produces structured reward specifications via a Planner agent, turns them into modular executable rewards, and then iteratively refines them through LLM-guided evolutionary operations (selection + crossover) under real environment feedback. A reward pool acts as specialized memory and a depth metric captures the structural complexity of rewards. Experiments on Tetris, Snake, Flappy Bird, and MuJoCo Humanoid show that FORGE consistently outperforms Eureka and REvolve.

**Strengths:**

- Clever decomposition into planner-based initialization + evolutionary refinement makes the framework both stable (at the beginning) and exploratory (later).

- Reward pool + depth is a lightweight and straightforward design to get interpretability and to analyze how complexity correlates with performance.

- Empirical results (3 games + 1 continuous control) show gains over strong recent baselines (Eureka, REvolve).

**Weaknesses:**

- The most important issue is the computational efficiency. Since each evolutionary step requires training/evaluating an RL policy under a new reward, Even if token usage is controlled, the overall wall-clock/sample cost can still be the main blocker for practical use. A comparison on “environment steps per performance gain” against Eureka/REvolve is missing.

- The method lacks ablations on different LLMs. The method assumes the LLM can both interpret two reward codes and synthesize a valid, environment-compatible offspring. It is unclear how robust this is to weaker models or higher code error rates. Some ablations with a smaller/older LLM, or with constrained generation, would clarify the robustness.

- The setup says all baselines are re-implemented under the same foundation model, but the paper does not detail how much prompt engineering/tuning effort was spent on Eureka/REvolve. Since the key claimed improvement is “structured initialization + evolution,” it would be good to show that Eureka/REvolve do not simply benefit from the same structured spec.

**Questions:**

Considering most modern LLMs have good vision capabilities, would feeding some vision information (e.g., game environments / failure trajectories) into the LLM helps further improve the sampling efficiency?

---

> ### Author Response · Authors · 2025-11-20
>
> Thank you for your comments and valuable advices! We address your concerns in the following.
>
> ---
>
> **P1:** The most important issue is the computational efficiency. The overall wall-clock/sample cost can be the main blocker for practical use, and a comparison on “environment steps per performance gain” against Eureka/REvolve is missing.
>
> **A1:** To address computational efficiency, we identify the limiting factors as i) Computational resources, ii) test environment and iii) RL algorithm. An intuitive way to demonstrate these effects is to draw comparison from REvolve paper:
>
> | |Resources|Environment|Algorithm| Clock Time per Generation|
> | ----------- | :-----------: |:-----------:| :-----------:| :-----------:|
> |Forge  | 16 Core Intel Emerald Rapids CPUs|Humanoid (5M steps)|PPO/SAC| 1.8 hrs/24hrs|
> |REvolve  |  16 NVIDIA A100 GPUs|Humanoid (5M steps) and Adroit Hand|SAC| 24 hrs|
>
> Other experiments on game environments can be completed within 1.5 hrs (per generation).
>
> We believe computational constraints largely arise from choices of engineering. For our purposes, we have to accommodate limiting resources but still be able to conduct meaningful experiments. Using only CPUs, we were able to do so relatively quickly, though at the cost of not abling to test on more complex environments. The generalizability still holds because we have showed that even latest LLM can struggle on simple game environments on the task of reward-shaping, and the performance differs under different frameworks.
>
> To compare "environment steps per performance gain" against baselines, we provide an analysis over the performance of trained RL policies. For each reward function $R$, the average performance gain for policy $\pi_R$ trained using $R$ is calculated as $\frac{1}{N-1}\sum_{i}(J(\pi^{(i+1)}_R)-J(\pi^i_R))$, where $i$ denotes the index of evaluation rollouts across N evaluations during training, and J is the cumulative rewards obtained in the corresponding rollout. This performance gain is then averaged across the generated reward functions in all generation rounds, except for the initialization round:
> | |Humanoid|Tetris|Flappy_bird| Snake|
> | ----------- | :-----------: |:-----------:| :-----------:| :-----------:|
> |Forge  | 11.653|0.258|1.485| 0.600|
> |REvolve  | 9.415|0.165|1.167| 0.359|
> |Eureka | 4.931|0.072|0.944| 0.342|
>
> The above results demonstrate that the RL policies trained using reward functions generated by Forge improve faster during training. We will add these experimental results in our updated version.
>
> ---
>
> **P2:** The method lacks ablations on different LLMs. The method assumes the LLM can both interpret two reward codes and synthesize a valid, environment-compatible offspring. It is unclear how robust this is to weaker models or higher code error rates.
>
> **A2:**  Thank you for the suggestions. We use Llama-3.1-70B-Instruct-Turbo and conducted  the same experiments, with result shown in the common questions section. We choose Llama-3.1 for its open-sourced property, release date (July 2024, compared to Claude-4-Sonnet, which was released in May 2025) and weaker benchmarks performance [1].
>
> Based on the experimental results, Forge still discovers better reward functions across all tested environments. The high success rates attributed to Forge and REvolve could be due to them performing crossover between two functions, which is a simpler task compared to mutating existing function. Eureka only mutates function without performing crossover.
>
> ---
>
> **P3:** Since the key claimed improvement is “structured initialization + evolution,” it would be good to show that Eureka/REvolve do not simply benefit from the same structured spec.
>
> **A3:** While Table 2 has demonstrated the effectiveness of our method, we provide a brief analysis on prompt engineering differences between REvolve/Eureka/Forge:
>
> REvolve and Eureka: Abstracted environment code + task decription
> Forge: Environment description + task decription + function signature
>
> One key difference is that Forge only provide textual description of the environment and task in the planning phase to facilitate more explorative planning. When generating code, Forge then specify the planned reward description and the exact function signature (parameters, return value). An experimental comparison would be to compare the performance of generated function in the initialization round, which is shown in Table 2, column zero-shot.
>
> ---
>
> **Q:** Considering most modern LLMs have good vision capabilities, would feeding some vision information into the LLM helps further improve the sampling efficiency?
>
> **A:** Thank you for the insight! We share the same vision that sample efficiency can be improved with LLM's multimodal capability, and we believe this would be a promising future direction.
>
> [1] “Llama 3.1 Instruct 70B: Intelligence, Performance & Price Analysis.” ArtificialAnalysis.ai,
> https://artificialanalysis.ai/models/llama-3-1-instruct-70b.

---

> > ### Author Response · Authors · 2025-11-27
> >
> > Dear Reviewer,
> >
> > Since the deadline is approaching, we are wondering if there are any additional details we can provide to address your concerns. We will gladly answer any further questions if needed.
> >
> > Thank you again for your time and we are looking forward to your response.

---

### Author Response · Authors · 2025-11-20
**Common questions**

In response to questions about robustness of our method to weaker/smaller and open-sourced model, we provide the experiment results using **Llama-3.1-70B-Instruct-Turbo** as the foundation model:
| |Humanoid|Tetris|Flappy_bird| Snake|
| ----------- | :-----------: |:-----------:| :-----------:| :-----------:|
|Forge  | 812.3|12.0|107.2| 19.4|
|REvolve  | 760.7|9.2|106.8| 17.6|
|Eureka | 702.3|4.2|16.4| 1.2|

Similar to Table 2, the above results report the cumulative rewards of the best-performing reward function generated after evolution.

The **success rates** (rates of successfully running generated functions) are summarized as follow:
Claude-4-Sonnet as foundation model:
| |Humanoid|Tetris|Flappy_bird| Snake|
| ----------- | :-----------: |:-----------:| :-----------:| :-----------:|
|Forge  | 0.98|0.99|0.99| 1.00|
|REvolve  | 0.94|0.99|1.00| 1.00|
|Eureka | 0.89|0.95|0.96| 0.93|

Llama-3.1-70B-Instruct-Turbo as foundation model:
| |Humanoid|Tetris|Flappy_bird| Snake|
| ----------- | :-----------: |:-----------:| :-----------:| :-----------:|
|Forge  | 0.91|0.94|0.93| 0.96|
|REvolve  | 0.94|0.99|1.0| 0.87|
|Eureka | 0.83|0.7|1.0| 0.64|

---

### Note · Authors · 2025-11-30

I have read and agree with the venue's withdrawal policy on behalf of myself and my co-authors.